# Foliar and Root Comparative Metabolomics and Phenolic Profiling of Micro-Tom Tomato (*Solanum lycopersicum* L.) Plants Associated with a Gene Expression Analysis in Response to Short Daily UV Treatments

**DOI:** 10.3390/plants11141829

**Published:** 2022-07-12

**Authors:** Alessia Mannucci, Marco Santin, Lucas Vanhaelewyn, Maria Calogera Sciampagna, Maria Begoña Miras-Moreno, Leilei Zhang, Luigi Lucini, Mike Frank Quartacci, Dominique Van Der Straeten, Antonella Castagna, Annamaria Ranieri

**Affiliations:** 1Department of Agriculture, Food and Environment, University of Pisa, 56124 Pisa, Italy; alessia.mannucci@agr.unipi.it (A.M.); maria.sciampagna@phd.unipi.it (M.C.S.); mike.frank.quartacci@unipi.it (M.F.Q.); antonella.castagna@unipi.it (A.C.); anna.maria.ranieri@unipi.it (A.R.); 2Laboratory of Functional Plant Biology, Department of Biology, Faculty of Sciences, Ghent University, 9000 Ghent, Belgium; lucas.vanhaelewyn@derooseplants.com (L.V.); dominique.vanderstraeten@ugent.be (D.V.D.S.); 3Department of Agricultural Economics, Ghent University, Coupure Links 653, 9000 Ghent, Belgium; 4Deroose Plants NV, Weststraat 129A, 9940 Evergem, Belgium; 5Department for Sustainable Food Process, Università Cattolica del Sacro Cuore, 29122 Piacenza, Italy; mariabegona.mirasmoreno@unicatt.it (M.B.M.-M.); leilei.zhang@unicatt.it (L.Z.); luigi.lucini@unicatt.it (L.L.)

**Keywords:** untargeted metabolomics, UHPLC-ESI-QTOF-MS, secondary metabolism, plant organs, qRT-PCR, ultraviolet radiation

## Abstract

Tomato (*Solanum lycopersicum* L.) is globally recognised as a high-value crop both for commercial profit and nutritional benefits. In contrast to the extensive data regarding the changes in the metabolism of tomato fruit exposed to UV radiation, less is known about the foliar and root metabolome. Using an untargeted metabolomic approach through UHPLC-ESI-QTOF-MS analysis, we detected thousands of metabolites in the leaves (3000) and roots (2800) of Micro-Tom tomato plants exposed to 11 days of short daily UV radiation, applied only on the aboveground organs. Multivariate statistical analysis, such as OPLS-DA and volcano, were performed to allow a better understanding of the modifications caused by the treatment. Based on the unexpected modulation to the secondary metabolism, especially the phenylpropanoid pathway, of which compounds were down and up accumulated respectively in leaves and roots of treated plants, a phenolic profiling was carried out for both organs. The phenolic profile was associated with a gene expression analysis to check the transcription trend of genes involved in the UVR8 signalling pathway and the early steps of the phenolic biosynthesis. The retention of the modifications at metabolic and phenolic levels was also investigated 3 days after the UV treatment, showing a prolonged effect on the modulation once the UV treatment had ceased.

## 1. Introduction

UV-A and part of UV-B radiation (UV, ultraviolet radiation; 290–400 nm) reach the Earth’s surface, triggering the photomorphogenic responses required for the acclimation of plants to this environmental stimulus [1,2,3]. Several studies have elucidated the role of UV radiation as a plant growth regulator; however, if exposed at a high dose, UV can negatively affect many physiological processes, causing the onset of oxidative stress that can no longer be counteracted by the boost of the antioxidant production, which is also triggered by UV light [4]. However, plants have developed several ways to cope with UV radiation, such as modifications of the metabolic pathways, along with an increase in the phenylpropanoid biosynthesis, which is one of the main topics under this environmental condition. Several other pathways are known to be responsive to UV radiation, including amino acid biosynthesis [5], as they also are precursors of specific secondary metabolites needed in the UV-B responses.

UV RESISTANCE LOCUS 8 (UVR8) has been identified as the specific photoreceptor able to perceive UV-B and trigger downstream signalling [6]. This dimeric protein is localized in both the nucleus and cytoplasm under normal light conditions; however, after UV-B exposure, it splits into monomers and binds to several light-regulating proteins such as CONSTITUTIVELY PHOTOMORPHOGENIC 1 (COP1) [7,8], leading to its accumulation in the nucleus compartment. The sequestration of COP1 inhibits the proteolysis of target proteins [1,8], among which is the transcription factor HY5 (ELONGATED HYPOCOTYL 5), enabling the transcription of downstream genes such as phenolic-related genes [4].

It has been widely demonstrated that metabolites such as phenolics and flavonoids are fundamental as protectants against oxidative damages caused by reactive oxygen species (ROSs) produced during stress responses, and they also act as UV-B-absorbing compounds in the epidermal cell layers, reducing the transmission of this wavelength to the mesophyll, and hence protecting the photosystems [2,9,10]. UV-B induces the transcription of genes encoding the enzymes involved in the first steps of the phenylpropanoid pathway, such as PHENYLALANINE AMMONIA LYASE (PAL), CHALCONE SYNTHASE (CHS), and FLAVONOL SYNTHASE (FLS) [11,12]. Therefore, the activation of the UVR8 signalling pathway enables plant morphologic modifications and the accumulation of UV screening compounds that enhance the plant’s antioxidant defence [6,13]. Notably, roots also express the UVR8 photoreceptor and several regulators required for UV-B signal transduction with some root specific gene induction [6,14,15]. Moreover, this wavelength is known to influence root development and direction of growth [16,17]; however, less is known regarding the biochemical changes occurring in this organ when only the shoot is exposed to UV-B.

This study aimed at having a collective view of the foliar and root metabolic modifications, with a focus on the phenolic profiling, during the acclimation of Micro-Tom tomato plants to short daily UV treatments for 11 days exclusively on the aboveground organs. Subsequently, the retention of the modifications at metabolic and phenolic levels was investigated at 3 days after the removal of UV treatment (recovery phase). Moreover, the quantification of phenolics with an “-omic” approach was associated with a gene expression analysis to check the transcription of genes involved in the UVR8 signalling and the early steps of the phenolic biosynthesis. Literature in which a metabolomic approach was applied to investigate UV and particularly UV-B effects on plants is still scarce, and, to the best of our knowledge, they were mainly addressed to fruit and foliar metabolome [18,19,20,21]. However, we know that roots are also equipped with the whole signalling molecules needed for UV responses [15,22], and that several signalling factors can generally move from shoot-to-root and vice versa [23]. Although light piping mechanism has been observed regarding far red light [24], similar evidence has not been reported for UV-B radiation yet. While the potential of UV radiation in increasing the plant antioxidant defence is well documented, here, we provide new insights regarding how UV exposure in the aerial part of the plant can affect below-ground tissues and how phenolics are differently modulated in the shoots, even after several days of recovery.

## 2. Results and Discussion

### 2.1. Effects of Short Daily UV Treatment of the Phyllosphere on the Leaf and Root Metabolism of Tomato Plants

The UHPLCESI/QTOF-MS analysis, coupled with a comprehensive database for primary and secondary metabolites identification (PlantCyc), allowed for the detection of about 3000 metabolites in leaves and 2800 metabolites in roots. The whole list of metabolites detected is provided as Appendix A, together with abundances and composite mass spectra (Appendix A). Given the broad chemical diversity in the metabolome, several multivariate statistical analyses were performed to allow a better understanding of the modifications caused by the UV treatment.

The output of the unsupervised fold-change-based hierarchical clustering (HCA) performed on the foliar metabolomic profile (Figure 1) showed a separation between UV and CTR samples, with the main clustering between UV-rec and the other groups. However, in the case of roots, the output of the HCA (Figure 1) showed the main clustering between CTR-rec and the other samples, with UV groups being clustered together irrespective of the time point considered. This preliminary analysis highlighted a different behavior of the two organs’ metabolism.

Then, an OPLS-DA supervised model (Figure 2a,b for leaves, Figure 2c,d for roots) was used to enhance the interpretation of the variables that maximize the differences among sample groups for each investigated organ.

Figure 2a,c clearly empathized the difference existing between CTR and UV samples (independently from the time point considered) in both leaves and roots. Consequently, Variable Importance for Prediction (VIP) analysis was carried out to show which metabolites weighed the most in the loading plot of these supervised models. Table 1 lists the metabolic markers in leaves, with 122 compounds in total, related to Figure 2a, with a VIP score higher than 1.1.

Univariate statistical analysis was performed to identify the VIP markers displaying a significant fold-change (FC) between UV and CTR foliar samples. About half of the compounds listed displayed a significant *p* value. Some compounds such as 2-oxoadipate, oxalosuccinate, scopolamine, 9-[6(S),9-diamino-5,6,7,8,9-pentadeoxy-β-D-ribo-nonafuranosyl]-9H-purin-6-amine, 3-hydroxy-9-apo-δ-caroten-9-one, and tricoumaroyl spermidine showed a remarkable and statistically significant enhanced FC in respect to CTR samples. On the contrary, N-P-tosyl-L-phenylalanyl chloromethyl ketone, 1-[18-hydroxyoeoyl]-2-[18-hydroxy-linoleoyl]-sn-glycerol, (22R,23R)-28-homocastasterone, and cyclo-dopa 5-O-glucoside reported a marked and significantly reduced FC in respect to CTR samples.

Roots VIP metabolic markers related to the OPLS-DA plot are shown in Figure 2c and listed in Table 2.

As for leaves, univariate statistical analysis led to identifying the compounds with a significant FC in respect to CTR root samples. Among 169 markers, 62 compounds displayed a significant FC. In particular, 10-deacetyl-2-debenzoylbaccatin III, a pentose, N-methylanthranilate, L-α-(methylenecyclopropyl)-glycine, 5,10-methylenetetrahydropteroyl mono-L-glutamate, all-trans-hexaprenyl diphosphate, 4-(β-D-glucosyloxy) benzoate, L-nicotianamine, dTMP, and N-(4-aminobenzoyl)-L-glutamate showed a remarkably enhanced or reduced (cycloheptadienyl/sinapate) FC in respect to CTR samples.

Finally, among the hundreds of VIP markers resulting from the OPLSDA plots, only eight markers were found to be in common between leaves and roots, and these were compounds related to the secondary metabolism such as O-sinapoylglucarolactone, cyanidin 3-O-β-D-p-coumaroylglucoside/cyanidin 3-(p-coumaroyl) glucoside/pelargonidin 3-O-β-D-caffeoylglucoside, scopolamine, aminoacids such as cyclogutamate and N-methylanthranilate, and 5-[[4-methoxy-3-(phenylmethoxy)phenyl]methyl]-2,4-pyrimidinediamine.

The OPLS-DA in Figure 2b,d shows the effects of UV and time factors in leaves and roots, respectively. A good separation among the different groups is visible in both organs; however, in leaves, the treatment played the main role for the distribution along the first vector, with UV samples placed in the negative half of the plot and CTR samples in the positive half, while the sampling time did not have a great effect. Surprisingly roots exhibited not only the effect of the UV treatment, with UV samples and CTR samples in the positive and negative half of the score plot considering the second vector, but also a time effect, with a good separation of the samples along the first vector according to this factor.

### 2.2. Outputs of the Volcano Analysis on the Metabolome

Differential compounds, with individual *p* values and fold-change resulting from the volcano plot, are fully provided as Appendix A. To visualise how the metabolic pathways were modulated by short daily UV treatments, the differential compounds resulting from the volcano analysis were interpreted by the Omics Viewer Dashboard of the PlantCyc Pathway Tool software. Figure 3 shows several metabolic pathways in leaves and roots, with the main focus on biosynthesis and degradation processes.

Leaves of UV-11d plants displayed an increased fold-change in the amino acid and hormone biosynthesis (+28 and +23 log_2_FC, respectively) in respect to the relative CTR, while the fatty acid/lipid and secondary metabolites showed a decrease in respect to CTR leaves on the same day (−24 and −100 log_2_FC, respectively, Figure 3). Especially the latter biosynthetic process was still reduced after the recovery, while fatty acid/lipid biosynthesis increased in respect to CTR (+47 log_2_FC). Fatty acids, as components of the cellular membrane, also play a role in stress signalling responses leading to the alteration in the fluidity of the membrane. As an example, the release of unsaturated fatty acids is able to activate the defensive genes in tomato leaves under herbivore wounding [25]. Yang and collaborators [26], studying the effects of high UV-B irradiation (120.8 μW cm^−2^ for 5 h followed by incubation with dark) on *Clematis terniflora*, found an accumulation of linolenic and linoleic acids. In our experiment, the fluctuation in the FA/lipid synthesis reported in UV-treated leaves and the increase in their biosynthesis after 3 days of recovery suggest a remodelling of the membrane in respect to untreated plants and once the treatment was removed. Specifically, the lipids assigned to the biosynthetic process comprised, among others, two galactolipids, one phospholipid, three sterols, and three polyunsaturated fatty acids. Among these latter, (12Z,15Z)-9,10-epoxyoctadeca-12,15-dienoate and (9Z,12Z)-15,16-dihydroxyoctadeca-9,12-dienoate, which are involved in the production of hydroxy fatty acids from α-linolenic acid, were oppositely affected by the UV-B exposure, being increased or decreased, respectively. Since some hydroxylated fatty acids act as antifungal agents [27], the influence of UV-B radiation on this pathway may lead to changes in plant–pathogen interactions.

Two of the three sterols identified as differential compounds according to volcano analysis (4α-formyl-4β-methyl-5α-cholesta-8,24-dien-3β-ol and 4α-carboxy-5α-cholesta-8,24-dien-3β-ol) markedly accumulated in UV-B-treated leaves after three days of recovery, similarly to the triterpene precursor, presqualene diphosphate (Appendix A). The two sterols identified in the present research are cholesterol precursors. Differently from mammals, who synthesize cholesterol, plants produce a complex mix of sterols, among which sitosterol, stigmasterol, and campesterol are the most representative compounds, though, in some plants, as in the Solanaceae family, cholesterol is present in significant amounts [28]. Sterols are intrinsic components of the cell membrane, therefore their increase after the end of UV-B treatment may represent a mechanism of repairing possible damage induced by UV-B radiation to restore membrane fluidity and permeability. A slightly increased lipid peroxidation was indeed observed in UV-B-irradiated Micro-Tom leaves on day 11, while, at the end of the recovery period, this parameter was similar between control and treated leaves [29]. It is well known that flavonoids can protect against lipid peroxidation caused by stressing conditions such as photoinhibition [30,31], and different studies have indicated that UV-B can increase lipid peroxidation levels, usually linked with a decrease in chlorophylls concentration and photosynthesis inhibition [32]. In particular, Liu et al. [33] found a significant correlation between anthocyanidins and lipid peroxidation, suggesting that the content of those antioxidant compounds may be linked to the extent of lipid oxidation. In this study, the FC of anthocyanin in leaves was slightly decreased in respect to CTR plants (paragraph 2.4), which is in line with the quantification reported in our recent paper [34] after 11 days. One could assume that flavonoids might have been consumed in reactions aimed to interrupt the oxidative process occurring at the expense of cellular components such as membranes. However, as we reported in our previous papers [29,34], the UV-B treatment applied in this study did not cause severe damages to the photosynthetic apparatus, as indicated by the higher total chlorophyll concentration detected after 11 days in treated plants, and no differences in respect to CTR, considering the PSII efficiency, were detected. Moreover, as showed in [34], we found a reduced expression of a gene related to anthocyanin biosynthesis on previous days, suggesting a likely reduced biosynthesis of these compounds in the leaves of treated plants.

An increase in sterol (sitosterol and stigmasterol) accumulation is reported in grapevine leaves following a ‘‘field-simulating’’ dose of UV-B radiation (4.75 kJ m^−2^ per day) administered at low intensity (16 h at 8.25 μW cm^−2^) and, less intensely, also at high intensity (4 h at 33 μW cm^−2^). This latter treatment induced an increase in antioxidant compounds, such as mono- and diterpenes and tocopherol and phytol [35]. In our study, the accumulation of some monoterpenes, diterpenes, and sesquiterpenes, among which were some phytoalexins, was stimulated in UV-exposed tomato leaves, indicating the activation of both MEP and mevalonate biosynthetic pathways. Such a response was generally more pronounced after 11 days of treatment (Appendix A). Beside changes in sterol levels, in our research, UV-B treated leaves underwent a decrease in 1-18:3-2-16:2-monogalactosyldiacylglycerol and an increase in the amount of 1-16:0-2-18:3-digalactosyldiacylglycerol, particularly after 3 days of recovery, indicating that UV-B radiation modifies the lipidic composition of membranes.

After 3 days of recovery, hormone degradation was strongly increased (+43 log_2_FC), suggesting the necessity of modulating the hormone homeostasis. Interestingly, in UV-11d leaves, the reduced secondary metabolites synthesis was parallel to a lower degradation process (Figure 3). A more detailed representation of the secondary metabolites pathway, which is the typical process involved in the UV response and acclimation, is shown in Appendix A, where, in particular, the phenylpropanoid biosynthetic and degradation pathways demonstrated a strong reduction in treated leaves.

Roots displayed a different behaviour from leaves, in particular, in the case of the secondary metabolism biosynthesis, which appeared to be enhanced in UV-11d in respect to its CTR (+191 log_2_FC, Figure 3). After examining the secondary metabolites in detail, pathway biosynthesis led us to identify an increase in terpenoid (+60 log_2_FC) and phenylpropanoid derivative (+16 log_2_FC) biosynthesis (Appendix A) in irradiated plants with respect to CTR at the 11-day treatment. However, phenylpropanoid biosynthesis was greatly reduced after the recovery (−47 log_2_FC). Degradative reactions of the secondary metabolites (Figure 3) showed a reduction in the fold-change after the recovery in respect to CTR roots (−13 log_2_FC). Concerning the carbohydrate metabolism, degradative processes were strongly enhanced on both day 11 of treatment and after the recovery (about +13 log_2_FC), with respect to roots of control plants, likely suggesting a mobilization of storage forms. The degradation of root carbohydrates might have been the source of precursors needed for the accumulation of secondary metabolites in this organ such as the flavonoid class, which will be discussed later in this article.

As observed in leaves, two monogalactosyldiacylglycerols showed an opposite behaviour following UV-B treatment also in roots. This finding, together with the decreased 4a-carboxy-5a-cholesta-7,24-dien-3b-ol root content detected in the roots of UV-B-treated plants, in accordance with the report of decreased cholesterol and accumulation of two steryl esters in roots of *Withania somnifera* plants irradiated with supplemental UV-B (3.6 kJ m^−2^ day^−1^ above ambient level [36], suggests a rearrangement of membrane lipid composition due to UV-B irradiation also in the non-irradiated organ.

Based on the unexpected modulation of the biochemical processes particularly related to the phenylpropanoid pathway, phenolic compounds were profiled in tomato leaves and roots, since it is highly recognised that phenolic compounds are involved in many plant processes and play a fundamental role in the acclimation towards UV, especially UV-B radiation.

### 2.3. Effects of Short Daily UV Treatment on the Phenolic Profile of Tomato Leaves and Roots

Since the UHPLC-ESI/QTOF-MS system coupled with a comprehensive phenolic database (Phenol-Explorer) allowed for the detection and identification of hundreds of compounds (232 and 183 compounds in leaves and roots, respectively), an OPLS-DA model was built for both organs, and for the metabolomics, to extrapolate the UV-induced phenolic modifications.

In leaves, the OPLS-DA analysis (Figure 4a) revealed a clear distinction between CTR and UV-treated groups, indicating that, regardless of the sampling time, the irradiation did result in effectively modulating the phenolic pattern.

Variable Importance for Prediction (VIP) analysis was carried out to show which phenolic compounds weighed the most in the loading plot of the OPLS-DA model reported in Figure 4a. Table 3 shows the 33 phenolic markers with a VIP score higher than 1.1, classified according to their class and subclass.

The most abundant class was the flavonoid one (15 compounds in total) followed by the phenolic acids (12), while anthocyanins (5), flavonols (5), and hydroxycinnamic acids (10) were the most represented subclasses. At the same time, to understand which of the VIP markers displayed a significant fold-change between UV and CTR foliar samples, a univariate statistical analysis was performed and highlighted three compounds with a significant *p* value. These are rosmarinic acid (−4 log_2_FC), caffeoyl aspartic acid (+0.52 log_2_FC), and cirsimaritin (+0.72 log_2_FC). Besides the UV protection of phenolic acids and flavonoids, they also play an important role as antifungal and antimicrobial compounds.

Furthermore, both sampling time and UV treatment played a role in the group separation along with the vectors (Figure 4b). Indeed, the first vector provided a clear separation of all groups, with CTR-11d being the only one located in the positive half of the plot, while, according to the second vector, only UV-11d was positioned in the positive half. However, upon obtaining a general overview of the OPLS-DA output, control, and UV-treated samples, they are shown to be well clustered within each group and well separated from the others, indicating good homogeneity among the replicates and a highly different phenolic profile according to both the treatment and the sampling time.

The OPLS-DA analysis on root tissue revealed a clear distinction between control and UV-treated plants (Figure 4c). This result suggests that the UV irradiation of the plant’s aerial part did indirectly alter the phenolic profile also within the below-ground tissues. The details of the VIP analysis performed on root tissue data are indicated in Table 4, where 25 phenolics with a VIP score higher than 1.1 are classified according to their class and subclass.

The most abundant class was the flavonoid one (12 compounds in total) followed by the phenolic acids (7), while anthocyanins (6) and hydroxycinnamic acids (5) represent the large subclasses. As for leaves, a univariate statistical analysis of the FC was performed, and p-Coumaroyl glycolic acid, a hydroxycinnamic acid, was the only compound with a significant *p* value. This phenolic acid was significantly reduced by UV treatment (−10 log_2_FC) and, as reported by Kadam et al. [37], it is known for its antioxidant and anti-inflammatory properties.

Moreover, considering both time and UV treatment factors (Figure 4d), a good separation among all groups occurred within the score plot. In detail, the first vector highlights the effect of the sampling time factor, since both CTR-11d and UV-11d plants were overlapped and placed in the negative half of the plot, while CTR-rec and UV-rec were in the positive half, although well distinct from each other. According to the second vector, the UV-rec group resulted in being the only one located in the positive half of the plot, distant from all the other groups.

### 2.4. Fold-Change Analysis on the Comprehensive Dataset of Leaves and Roots Phenolic Compounds

To investigate the accumulation of the most UV-responsive phenolic classes and subclasses in detail, a fold change (FC) analysis of UV in respect to controls was carried out on the base of the whole phenolics dataset, considering both the sampling times (11 days of UV exposure and 3 days after the end of the treatment) and both tissues (leaves and roots) (Figure 5). A detailed list of the phenolic molecules and their fold change are listed in Appendix A.

A general decrease in the phenolic class abundance was detected in the UV-11d leaves, especially for phenolic acids, which showed a −0.57 log_2_FC in respect to control leaves (Figure 5a). After the 3 days of recovery, most classes were still down accumulated in UV-rec, with flavonoids and phenolic acids showing a −0.13 and −0.18 log_2_FC and lignans being the most affected phenolic subclass, with −0.68 log_2_FC in respect to control (Figure 5a). UV-induced responses typically trigger the accumulation of phenylpropanoids due to their ROS scavenging and screening activity; although, recently, several studies have showed the complexity of the phenolic response to this wavelength, depending on the exposure time, plant species, and phenolic class considered [38]. However, after 11 days of treatments, Micro-Tom tomato plants have already acclimated to the supplemental UV irradiation, as showed in our previous study [29], where many physiological parameters were checked to ensure that the UV dose applied did not induce stress-related responses. Indeed, we detected a reduced content of total phenols on day 11 of UV (according to the Folin–Ciocalteau method), and no differences were found in terms of flavonoid concentration. In contrast, a slightly increased phenols and flavonoids content was detected in previous days. According to these data, we speculate that the well-known and widely described induction of phenolics biosynthesis could be an early response under UV-B conditions, with fluctuations depending on the antioxidant status of the plant. After the plant acclimated to low UV-B doses and the morphogenic responses took place, e.g., after an increase in leaf thickness, lignification, trichome density [39,40,41], it is likely that the plant energy and resources are re-addressed mainly to primary rather than secondary metabolism, so that physiological growth and development can be restored. Indeed, as reported by Coffey et al. [42], the inhibiting effect of UV-B radiation on the leaf growth of *A. thaliana*, grown in outdoor conditions for 7 months, ten days a month, was detected only under summer intensities, and it was not related to the activation of the UVR8-signalling pathway. The authors also found no changes in the concentration of UV-absorbing pigments; however, nothing was reported about the composition of the pool. Robson et al. [16] also argued that the UV effects on leaf growth are complex and that, once the plants acclimated to the UV stimulus, normal development can be restored, leading to compensatory effects on the enlargement of the leaf area. According to our hypothesis of a UV-triggered modification in the morphology of the leaves, the reduced levels of lignans detected in UV-rec leaves suggest that these compounds were cross-linked for lignin biosynthesis [43].

Very few studies have focused on UV-B perception in roots [15,44], and most of them have applied high UV-B intensities, which likely induced the onset of stress responses overlapping the activation of the UVR8 pathway [45,46]. To the best of our knowledge, no previous works have investigated the changes in the phenolic profile of roots under short daily UV conditions. The fold-change analysis on the root phenolic abundance dataset (Figure 5b) revealed a different behaviour between the two plant organs. The roots of treated plants showed an increase in flavonoids (+0.35 log_2_FC) and lignans (+0.26 log_2_FC) in the UV-11d group. After the recovery, an opposite behaviour was shown by lignans (−0.27 log_2_FC) and phenolic acids (−0.60-fold) whose concentrations were both reduced, while flavonoids were still increased (+0.64-fold).

Consequently, to better understand how the treatment influenced the flavonoids (Figure 5c,d) and phenolic acids (Figure 5e,f), the fold change analysis was also carried out at subclass levels in both organs.

Flavonoids in UV-11d leaves showed an increased log_2_FC of dihydrochalcones, flavanols, flavanones, and flavones (+0.25, +0.91, +1.47, +0.93, respectively), while anthocyanins, dihydroflavonols, and isoflavonoids were reduced (−0.16, −0.81 and −0.43, respectively) compared to the correspondent control. After the 3-day recovery, leaves of treated plants still exhibited an increase in flavanols and flavones (+0.44 and +0.22 log_2_FC, respectively) and a persistent decrease in anthocyanins and dihydroflavonols (−0.13 and −0.68, respectively). Conversely, flavanones (−0.22 log_2_FC), flavonols (−0.30), and isoflavonoids (+1.45) demonstrated an opposite behaviour in respect to what was observed after 11 days. In root tissue, all the flavonoid subclasses generally increased their abundance at both sampling times, with flavones showing the highest increase (+0.74 and +1.52 log_2_FC in UV-11d and UV-rec, respectively). Only isoflavonoids were reduced after the recovery period, with a −1.69-fold decrease in respect to controls.

Besides their function as antioxidants, flavonoids can modulate the phytohormone signalling as well. Polar auxin (IAA) transport was increased in studies with *Arabidopsis* mutant, with reduced levels of flavonoids aglycones [47,48,49]. In our study flavonols and flavones, which have been demonstrated to be able to inhibit polar IAA transport and signalling [50,51], were found to be enhanced in UV-11d leaves, likely causing a reduced basipetal movement of this hormone towards roots. Indeed, a reduced level of IAA concentrations was detected in the roots of UV-treated plants after 3 days of recovery [29], although a reduced biosynthesis in this organ could be another factor affecting its amount. Although less concentrated in respect to the aerial part of the plant, root flavonoids have significant roles in the regulation of root growth, within the nitrogen cycle, and among allelopathic interactions [52]. Flavonoids can exert signalling actions in lower concentrations than those necessary for ROS scavenging and UV absorption [50,53]. The increased levels of flavonols and flavones detected in roots of UV-11d may play a role in creating an IAA gradient at cellular and tissue levels, thus regulating root growth. However, since the flavonoid aglycones are located in tissue involved in auxin translocation, it is reasonable to consider that 8 flavonol glycosides out of 11 flavonols were found, and only 1 flavone glycoside out of 6 flavones in leaves, while, in roots, there were 6 flavonol glycosides out of 9 flavonols and 1 flavone glycoside out of 5 flavones. Further investigations are needed to verify the link between the IAA metabolism and flavonoids under short daily UV treatments, considering both leaves and roots, to gain a complete understanding of this phenomenon. The presence of flavonoids has been reported in the root exudates released into the rhizosphere, where these compounds act as regulators of nutrient availability, thanks to their reducing potential and chelating ability and promotion of root-rhizobia symbiosis [54]. Though we did not study the flavonoid content and composition of the root exudated, the increased flavonoid content (FC) in the roots of treated plants, detected in our experiment, suggests a potential benefit for irradiated plants in minerals uptake.

Concerning the phenolic acids, a general decrease in hydroxybenzoic (HB) and hydroxycinnamic (HC) acids occurred in leaves at both UV-11d and UV-rec. Notably, the strongest decrease was detected at 11 days for HC (−0.60 log_2_FC) and after the 3-day recovery period for HB (−1.43-fold). In a study on the effects of UV-modulated resistance to herbivory in tomato plants (0.34 kJ m^−2^ day^−1^ UV dose), it was found that chlorogenic acid and rutin, the two major leaf phenolic compounds in this species, but generally the whole leaf metabolome, did not change after 14-day UV exposure [55]. This might suggest that the secondary metabolism could have been affected at earlier days of the UV treatment. HC and HB in roots displayed an opposite trend, with −1.29 and +0.61 log_2_FC for HB and HC at 11 days, respectively, and +0.87 and −0.48 log_2_FC for HB and HC after the 3-day recovery period. As already mentioned, phenolic acids are involved in plant growth, antioxidant defence, and plant/microbe interaction [56,57]. Indeed, these compounds can be secreted by roots and act as allelochemicals by suppressing the proliferation of pathogens [58].

### 2.5. Outputs of the Volcano Analysis on the Phenolic Profile

A volcano analysis was performed on the phenolic dataset, combining moderated *t*-test (*p* < 0.05) and fold-change analysis (cut-off ≥ 1.2), and the differential compounds according to each sampling time were grouped in their respective class and subclass. The output revealed a similar trend between leaves and roots in the number of differentially modulated compounds (Table 5 and Table 6).

Indeed, a higher number of up and down accumulated compounds was found after the recovery, indicating that the short daily UV treatment applied for 11 days affected the plants not only during the daily irradiation, but also after 3 days of recovery.

Moreover, the effect on the UV-driven modulation of the phenolics differed between the two organs. Indeed, in leaves, 14 phenolics increased, while 10 phenolics decreased, following the UV treatment (Table 5). Considering the roots, only three phenolics increased, while 13 phenolics decreased, by UV exposure (Table 6).

### 2.6. Effects of Short UV Treatment on the Gene Expression of Tomato Leaves and Roots

To integrate the timing of response between the UV-mediated gene modulation and phenolics accumulation, leaves and roots were sampled at 8 d and 11 d of UV treatment and 3 days after the end of the UV irradiation (rec). For this purpose, some candidate genes related to the UVR8 specific pathway, HY5 and COP1, were investigated due to their well-known implication in the responses to low UV-B doses [4]. Simultaneously, genes related to the phenylpropanoid biosynthetic pathway, PAL, CHS, and FLS, were tested to find a link with the changes in the phenolic profile.

#### 2.6.1. Differences in the Gene Expression of UVR8 Pathway-Related Genes

Leaf HY5 and COP1 gene expression is reported in Figure 6a,c. As expected, the HY5 transcript level increased at both 8 d and 11 d days (+2.1- and +3.88-fold, respectively), while no differences compared to control plant levels were found three days after the end of the UV exposure. This trend confirmed the activation of the specific UVR8 pathway in Micro-Tom tomato plants by the application of short UV treatment. Indeed, HY5 is known to play a pivotal role in light signalling-mediated responses, and it is also known to be induced by UV-B radiation [4].

Considering the root tissue, a decrease in HY5 expression was observed in treated plants (Figure 6b), being statistically significant at 8 d of UV and after the recovery (−0.53 and −0.18-fold, respectively). To the best of our knowledge, the expression of HY5 in roots not directly exposed to short daily UV doses has never been investigated in tomatoes during a long-term experiment. Moreover, in general, studies on the roles and functions of HY5 in roots are scarce. HY5 is known to take part in many biological processes, such as nitrogen assimilation and other activities related to light and hormone control, as in the case of auxin [4,59]. In a study on Arabidopsis, Cluis et al. [59] found that two negative regulators of auxin signalling, AUXIN RESISTANT 2/INDOLE ACETIC ACID 7 and SOLITARY ROOT/IAA14, were downregulated in hy5 mutants, suggesting that HY5 might regulate their transcription by binding their promoters. In addition, the same authors stated that hy5 mutants have higher IAA signalling due to the HY5-regulating effect on the expression of those IAA inhibitors. From our data, we speculate that the reduced expression of the HY5 gene in the roots of UV-treated plants could potentially lead to changes in the IAA signalling and alterations in the morphology of this organ. Further studies regarding this aspect are encouraged.

No modification in COP1 gene expression was detected in both leaves and roots (Figure 6c,d), suggesting that its activation could have occurred in the early days of the treatment, or that it might require some recovery time after the treatment to be detectable. However, in accordance with our results, a study by Liu et al. [12] on the gene regulation of flavonol synthase by high and low UV-B in *Vitis vinifera* L. grapes found no significant responses COP1 to UV-B.

#### 2.6.2. Differences in the Gene Expression of Some Phenylpropanoid Biosynthetic Genes

The UV effects on the phenolic biosynthetic pathways were checked by measuring the gene expression of PAL, CHS, and FLS, whose enzymes are involved in the early steps of phenolic acid, flavonoids, and flavonols biosynthesis, respectively.

PAL, an enzyme catalysing the cinnamic acid biosynthesis, which is a precursor for lignin and flavonoids, plays a pivotal role in regulating compounds resulting from the phenylpropanoid pathway and as a branch point between aromatic amino acids and secondary metabolites pathway [60]. The transcripts levels of PAL were increased (+1.94-fold) in leaves of treated plants at 11 d (Figure 6e) in respect to controls, while no differences were found on the other days. This is in line with the differences in the fold-change of phenolic acids’ content, presented in Figure 5a, and which, despite being less concentrated than in the control leaves and being less decreased in UV-rec than in UV11d, indicate a trend to recover the control values, likely due to an upregulation of the related biosynthetic enzymes. In roots (Figure 6f), an increase in the PAL transcript level of treated plants was detected at 8 d (+2.76-fold), followed by a reduction in the latter days (−0.44 and −0.73-fold at 11 d and rec, respectively). The changes in PAL transcript level occurring in roots reflect the changes in the phenolic acids’ content (Figure 5b). Remarkably, the increase in PAL gene expression at 8 d led to an increase in the phenolic acids’ content in UV-11d, while the reduced concentration of these compounds after the 3 days of recovery matched the reduced PAL levels detected at 11 d. Moreover, a persistently reduced amount of root phenolic acids was likely to be detected in the later days, since the downregulation of PAL transcription was still visible after the end of the treatment.

CHS is an enzyme involved in the early steps of flavonoid biosynthesis, catalysing the conversion from coumaroyl-CoA to naringenin chalcone, while FLS catalyses the conversion from dihydroflavonoids to flavonoids. The literature states that the increase in CHS expression represents a UV-B-specific response, upregulated by very short UV-B exposures [61]. Moreover, Brown and Jenkins [4] have indicated that changes in CHS expression do not occur when non-specific UV-B responses take place. Foliar CHS transcript levels (Figure 6g) were reduced by UV treatment at both 8 d and 11 d (−0.65 and −0.69-fold, respectively) and, although the flavonoid content in UV-11d was comparable to controls (Figure 5a), lower CHS levels might be the cause of the reduction in the flavonoid amount detected in UV-rec (Figure 5a). After the recovery, CHS gene expression slightly increased (+0.19-fold) compared to control samples, suggesting the activation of the flavonoid biosynthesis and their accumulation in the following days. No differences were found in foliar FLS gene expression (Figure 6i) according to the days considered, even if a decreased flavonol accumulation was evident in UV-rec (Figure 5a).

In roots, CHS transcript levels were reduced (−0.65-fold) at the 8 d (Figure 6h), followed by a 3-fold increase at 11 d and rec, along with the increase in flavonoid concentration (Figure 5b). Root FLS gene expression (Figure 6j) exhibited an increase in UV-treated plants at 8 d (+0.58-fold), possibly leading to a slight increase in the flavonols detected in the UV-11d root samples (Figure 5b).

Generally, a good relationship between gene expression and biochemical analysis was provided. One of the main results worth highlighting is that that the UV irradiation applied on the plant’s aerial part caused the activation of some genes related to the phenolic biosynthetic pathway of the root. As mentioned before, in the case of UV, no light piping has been observed so far, contrary to far red light [24]; therefore, one might hypothesise that some kind of biochemical signals moved to the root organ to provide a coordination between the upper and lower ground parts of the plants.

Besides a local accumulation mediated by enhanced gene expression, a long-distance transport from leaves to root cannot be excluded. As demonstrated by Buer et al. [62] by the local application of a fluorescent flavonoid and its detection in distal tissues in *Arabidopsis*, flavonoids accumulated in light-shielded roots of plants with light-exposed shoots thanks to a shoot-to-root transport.

However, it should be remembered that the metabolites level results from both biosynthetic and degradation processes, and the overall decrease in phenolic acids and flavonoids in leaves could be also due to their consumption in ROS scavenging activity. Additional time points on the earlier days are highly encouraged to have a comprehensive view of the acclimation process of Micro-Tom tomato plants under short daily UV treatment.

## 3. Materials and Methods

### 3.1. Plant Cultivation and UV Treatment

Surface sterilized seeds of *Solanum lycopersicum* L. cultivar Micro-Tom (JustSeed Ltd., Wrexham, United Kingdom) were germinated on water-soaked paper and then transferred in pots with perlite for 7 days, after which the plantlets were placed in a hydroponic system and grown in a Hoagland solution (pH~6), refreshed every week. The climate chamber was set at 24 °C, with 16 h light/8 h dark photoperiod and a photosynthetic photon flux density (PPFD) of 228 μmol m^−2^ s^−1^ supplied by an LED illumination system (blue/red/green 3/6/1 ratio, C-LED, Imola, Italy). The UV treatment was applied after 24 days and lasted 11 days. Plantlets were divided into a control group (CTR), receiving PAR radiation only, and a UV-treated group (UV), grown with PAR + UV radiation (15 min a day corresponding to a daily unweighted UV dose of 1.19 kJ m^−2^) provided by Philips Ultraviolet-B Narrowband lamps (TL 20 W/01-RS, Koninklijke Philips Electronics, Eindhoven, The Netherlands). The UV bulbs emission, measured by a JAZ EL-XR1 spectroradiometer (OCEAN OPTICS, Dunedin, FL, USA), provided a UV-B biologically effective intensity of 0.132 W m^−2^ and UV-A intensity of 0.015 W m^−2^, calculated according to Flint and Caldwell [63], with the latter corresponding to only 36% contribution of the total UV emission of the lamps. Different shading between CTR and UV plants was avoided through the application of empty lamp holders inside the CTR chamber, and all plants were rotated within the chambers to obtain uniform irradiation.

For phenolic profiling, samples of leaves and roots of both control and treated groups were collected after 11 days of short daily UV treatment (CTR-11d and UV-11d) and after 3 days from the end of the irradiation (CTR-rec and UV-rec). For gene expression analysis, samples were collected after 8 (8 d) and 11 (11 d) days of UV exposure and after 3 days from the end of the irradiation (rec). A pool of leaves and the whole root were immediately frozen in liquid nitrogen and stored at −80 °C until use.

### 3.2. Extraction and UHPLC-ESI-QTOF-MS Analysis of Leaves and Roots Metabolites

Freeze-dried samples (100 mg) of leaves and roots were extracted by homogenizer-assisted solvent extraction (Ultra-Turrax; Polytron PT, Switzerland) in 4 mL of an acidified methanol/water solution (80:20, *v*/*v*), centrifuged and filtered through a 0.22 µm membrane. For the analysis, 4 biological replicates per treatment and time, each obtained from a pool of 3 different plants, were used. The analysis of the metabolomics and phenolic profile was performed as reported by Senizza et al. [64] by using an Agilent 6550 iFunnel quadrupole-time-of-flight mass spectrometer and an Agilent 1200 series ultra-high-pressure liquid chromatographic system (UHPLC-ESI/QTOF-MS). Reverse phase chromatographic separation was achieved under a water-acetonitrile gradient elution, starting from 6% acetonitrile to 94% in 33 min. The mass spectrometer worked in SCAN mode with a range from 100 to 1200 m/z, positive and negative polarity, and extended dynamic range mode (nominal mass resolution = 30,000 FWHM).

Metabolite annotation was performed by the Profinder B.07 software tool (Agilent Technologies, Santa Clara, CA, USA) by their monoisotopic accurate mass and isotopes patterns (accurate spacing and isotopes ratio) according to [64], using the databases Phenol-Explorer 3.6 and PlantCyc to annotate phenolic compounds and plant metabolome, respectively [65,66]. Identification was carried out in compliance with Level 2 (putatively annotated compounds), as set out by the COSMOS Metabolomics Standards Initiative [67].

### 3.3. RNA Extraction and qRT-PCR Analysis

RNA from frozen homogenized plant tissue (3 biological replicates per treatment and time, each obtained from a pool of 3 different plants per treatment and time) was extracted from leaves and roots separately and prepared using Qiagen RNeasy Plant Mini kits (QIAGEN S.r.l., Milan, Italy) according to the manufacturer’s instructions and followed by cDNA synthesis using the cDNA Verso kit (Thermo Fisher Scientific-Life Technologies, Carlsbad, CA, USA). The SyGreen qPCR kit (qPCRbio; PCR Biosystems, London, UK) was used to perform qRT-PCR in an iCycler thermal cycler with an iQ5 optical system (Bio-Rad, Hercules, CA, USA) as an optical module for signal detection. The data were analysed with QBase software (Biogazelle, Ghent, Belgium) [68], where expression values were normalised with reference genes ACTIN7, UBIQUITIN, and EF1. The relative expression was automatically calculated in Qbase^+^. Primer sequences for HY5, COP1, PAL, CHS, FLS, and reference genes are listed in Appendix A.

### 3.4. Data Elaboration and Statistics

Elaboration of metabolomics data was carried out as previously reported by Ghisoni et al. [69]. The software Mass Profiler Professional 12.6 (Agilent Technologies, Santa Clara, CA, USA) was used for log_2_ transformation, normalization at the 75th percentile, and baselining against the median of each compound. Thereafter, the unsupervised hierarchical cluster analysis was used to underline the relatedness across the different treatments, according to Euclidean distance and Ward’s linkage rule. The volcano plot analysis was also performed for each treatment by combining a moderated *t*-test (*p* < 0.05; Benjamini–Hochberg multiple testing correction) and fold-change analysis (cut-off ≥ 1.2). The SIMCA 16 (Umetrics, Malmo, Sweden) software was then used to build the orthogonal projection to latent structures discriminant analysis (OPLS-DA) supervised modelling. Outliers were investigated according to Hotelling’s T2 (95% and 99% confidence limits for suspect and strong outliers), CV-ANOVA (*p* < 0.05) cross-validation and permutation testing (for overfitting, N = 100) were also performed from the OPLS-DA model. Thereafter, model fitness parameters (goodness-of-fit, R2Y, and goodness-of-prediction, Q2Y) were recorded, and the variable importance in projection (VIP) analysis was used to identify the most discriminant compounds (VIP score > 1.1). Differential compounds were finally imported into the PlantCyc Omic Viewer Pathway Tool (Stanford, CA, USA) software for biochemical interpretations [70].

JMP software (SAS Institute, Inc., Cary, NC, USA) was used for gene expression data to perform Student *t*-test statistical analysis. According to each day (8 d, 11 d and rec), asterisks (*) indicate a significant difference between CTR and UV group (* *p* ≤ 0.05, ** *p* < 0.01, *** *p* < 0.001).

## 4. Conclusions

To the best of our knowledge, this is the first time that a comparative metabolomic and phenolic profiling analysis has been performed in leaves and roots following UV irradiation. In the present work, we focused on the late responses of tomato plants acclimated to short daily UV treatment. The untargeted approach revealed an extensive modulation of the metabolome in leaves. Nonetheless, the root metabolome was also significantly affected by UV, despite not being directly exposed to the radiation, thus giving a new insight on the indirect effects in this organ, indicative of shoot-to-root signalling. Our results demonstrate that both biosynthesis and degradation processes were affected, involving AA and FA/lipids, in addition to a modulation of the hormone profile in leaves as compared to non-irradiated plants. Moreover, roots displayed a marked increase in the carbohydrate degradative pathway. However, a particular trend related to the secondary metabolism was noticed in both organs. Indeed, the biosynthesis of secondary metabolites appeared reduced in the leaves of treated plants. This result may indicate that, once the plant had acclimated to short daily UV treatment, the compounds produced first as part of the antioxidant defence could be used for other purposes, e.g., cross-linked for lignin synthesis in the case of lignans, thus causing changes in the leaf morphology and development. Roots showed an opposite behavior, with an enhancement of the secondary metabolism, particularly after the 11 days of treatment, where an increase in flavonoid content was found. These changes might affect not only the nutrient uptake, but also the architecture of this organ. Further studies are encouraged to verify this hypothesis. Our data also suggest a prolonged effect on the modulation of phenolic compounds in both organs once the UV treatment had ceased.

## Figures and Tables

**Figure 1 plants-11-01829-f001:**
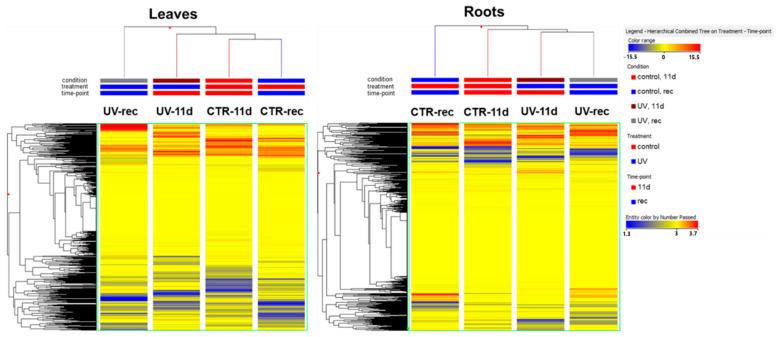
Unsupervised hierarchical clustering analysis of the Micro-Tom tomato metabolome in leaves and roots clustered considering the treatment (control, CTR, or UV-treated, UV) and time (11 days of UV, 11 d, or 3 days after the end of the UV exposure, rec). A fold-change-based heat map was used to create dendrograms. Clustering and dendrograms were produced by choosing the Euclidean distance and Ward’s linkage rule.

**Figure 2 plants-11-01829-f002:**
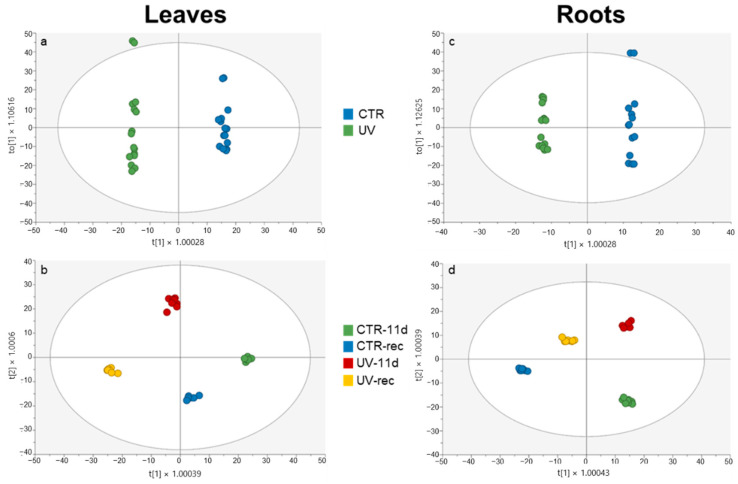
OPLS-DA carried out from the UHPLC-ESI-QTOF metabolome of leaves and roots of Micro-Tom tomato plants, considering (**a**,**c**) the UV treatment factor; (**b**,**d**) UV treatment and time factors. Individual replications are given in the class prediction model score plot.

**Figure 3 plants-11-01829-f003:**
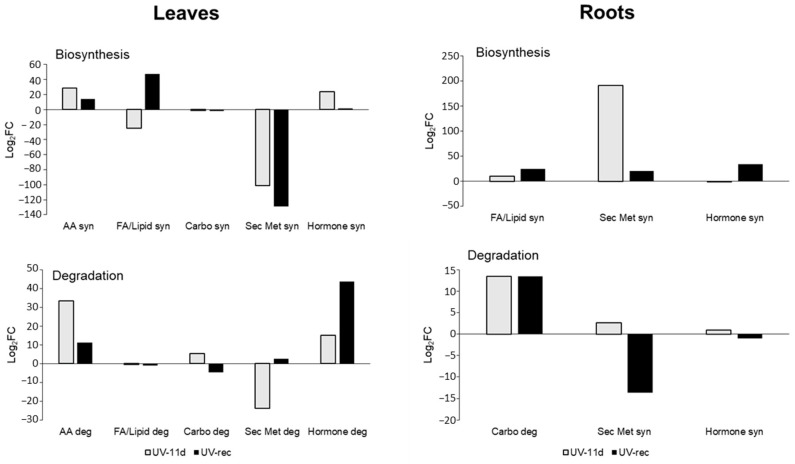
Fold-change (FC) of metabolic processes in leaves and roots of UV treated plants, at 11 days of treatment and after 3 days of recovery, with respect to control plants. FC values were elaborated by the Omics Viewer Dashboard of the PlantCyc Pathway Tool software. Data for each biochemical pathway represent the sum of Log_2_FC of each detected compound. Syn, biosynthesis; Deg, degradation; FA/Lip, fatty acids and lipids; Carbo, carbohydrates; Sec Metab, secondary metabolites; AA, amino acids.

**Figure 4 plants-11-01829-f004:**
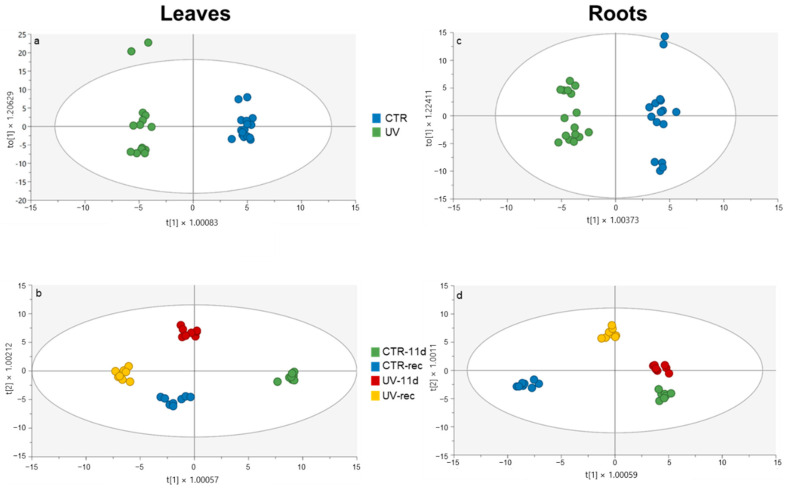
OPLS-DA carried out from the UHPLC-ESI-QTOF phenolic profile of leaves and roots of Micro-Tom tomato plants, considering (**a**,**c**) UV treatment (**b**,**d**) UV treatment and time factors. Individual replications are given in the class prediction model score plot.

**Figure 5 plants-11-01829-f005:**
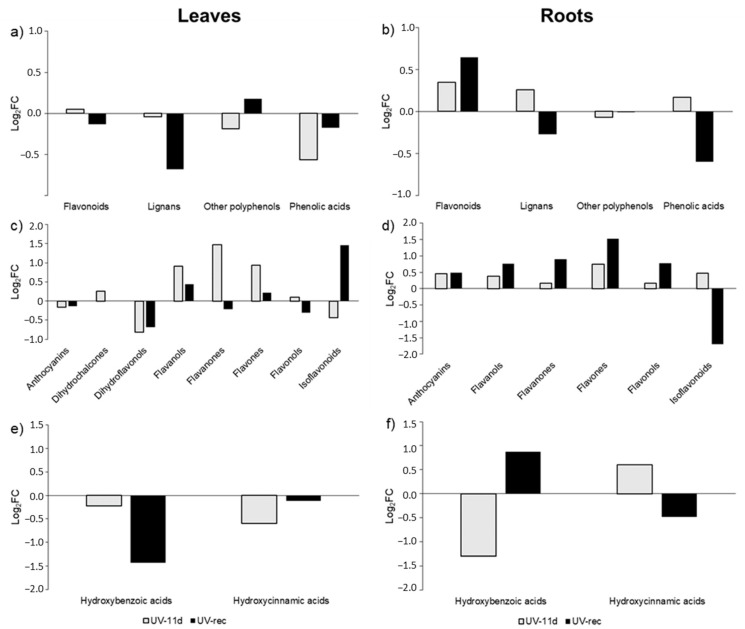
Fold-change (FC) of phenolic accumulation in leaves and roots of UV treated plants at 11 days of treatment and after 3 days of recovery with respect to control plants. Compounds were grouped on the basis of phenolic class (**a**,**b**), flavonoid (**c**,**d**), and phenolic acids (**e**,**f**) subclasses. Data for each biochemical pathway represent the sum of the Log_2_FC of each detected compound.

**Figure 6 plants-11-01829-f006:**
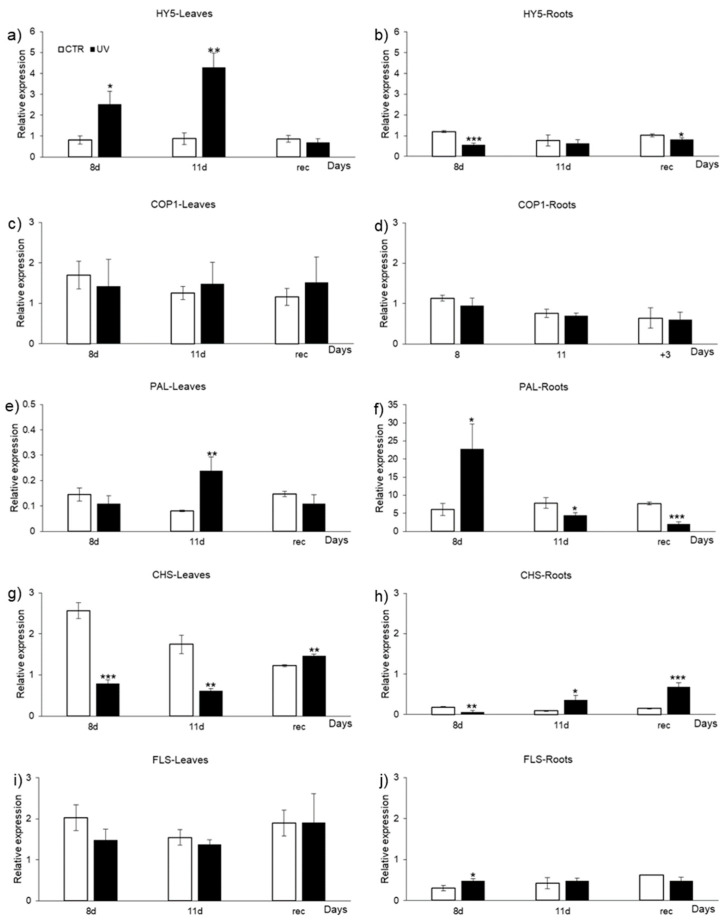
Expression levels of genes involved and in the UVR8 pathway (HY5, COP1) and in phenolic biosynthesis (PAL, CHS, FLS) in (**a**,**c**,**e**,**g**,**i**) leaves and (**b**,**d**,**f**,**h**,**j**) roots of control (CTR) and UV-treated (UV) Micro-Tom tomato plants after 8 and 11 days of UV treatment (8 d and 11 d) and after 3 days from the end of the irradiation (rec). Data are mean ± SE of 3 biological replicates. According to each day, asterisks (*) indicate a significant difference between CTR and UV group (* *p* ≤ 0.05, ** *p* < 0.01, *** *p* < 0.001).

**Table 1 plants-11-01829-t001:** Leaf metabolic markers, according to their VIP score ± standard deviation (STD), related to the OPLS-DA model in Figure 2a (effect of UV treatment). Compounds are provided with *p* value and their Log_2_FC calculated with respect to control samples. * *p* ≤ 0.05, ** *p* < 0.01, *** *p* < 0.001, n.s., not significant.

Compound	VIP Score ± STD	*p* Value	Log_2_ FC
2-oxoadipate	2.23 ± 0.51	***	14.39
2,4-pyridinedicarboxylate/quinolinate	2.09 ± 0.41	***	−0.58
rosmarinate	2.07 ± 0.54	n.s.	0.34
desmethylxanthohumol	2.01 ± 0.43	n.s.	−10.09
gibberellin A	1.98 ± 0.40	***	−0.62
phlormethylbutanophenone	1.96 ± 0.54	n.s.	4.00
O-sinapoylglucarolactone	1.95 ± 0.89	***	−0.64
(-)-epipodophyllotoxin/β-peltatin/podophyllotoxin/5′-demethoxy-6-methoxypodophyllotoxin	1.94 ± 0.55	***	0.52
L-α-amino-ε-keto-pimelate	1.92 ± 0.54	n.s.	−8.37
ubiquinone-2	1.92 ± 0.50	n.s.	5.06
α-difluoromethylarginine	1.91 ± 0.76	n.s.	−6.03
oxalosuccinate	1.90 ± 0.86	***	12.36
1-O-(4-coumaroyl)-β-D-glucose	1.88 ± 0.37	n.s.	13.85
(E)-4-hydroxy-3-methylbut-2-en-1-yl diphosphate	1.87 ± 0.65	***	−0.31
3-deoxy-D-manno-octulosonate 8-phosphate	1.86 ± 0.88	***	−0.57
pelargonidin/genistein/emodin/apigenin	1.84 ± 0.54	n.s.	−10.19
4,5-seco-dopa	1.83 ± 0.51	n.s.	−10.18
sn-glycero-3-phosphocholine	1.83 ± 0.61	n.s.	13.75
N-methylanthranilate	1.82 ± 0.36	***	−0.67
(+)-pisatin/1,7-dihydroxy-6,8-dimethoxy-2-methylanthraquinone/robustaquinone B/dimethylkaempferol/2′,7-dihydroxy-4′,5′-dimethoxyisoflavone/cirsimaritin/ladanein	1.82 ± 0.83	***	0.51
solasodine	1.80 ± 0.91	**	0.78
S-adenosyl-4-methylsulfanyl-2-oxobutanoate	1.79 ± 0.76	n.s.	n.s.
2′-deoxymugineate	1.78 ± 0.37	*	−0.26
2-hydroxy-4-carboxypyrimidine	1.75 ± 0.77	n.s.	−10.39
3-C-glucosyl-2,4,4′,6-tetrahydroxydibenzoylmethane/6-C-glucosyl-2-hydroxynaringenin	1.74 ± 0.70	n.s.	n.s.
A pentose	1.72 ± 0.50	***	−0.72
acetonedicarboxylate/2-oxoglutarate	1.72 ± 0.89	*	−0.46
5-methylcytidine/N4-aminocytidine	1.71 ± 0.81	n.s.	14.58
2-deoxyglucose 6-phosphate/β-L-fucose 1-phosphate/	1.69 ± 0.46	n.s.	−12.70
methylsuccinate/glutarate	1.69 ± 0.98	**	−1.62
pyridoxal	1.69 ± 0.70	n.s.	−10.19
(indol-3-yl)pyruvate	1.69 ± 0.99	**	−0.62
tetraketide pyrone	1.69 ± 0.74	n.s.	n.s.
capsidiol/lubimin	1.68 ± 0.58	*	0.29
coniferaldehyde	1.68 ± 0.37	n.s.	11.10
arabidopate	1.67 ± 0.64	n.s.	0.34
dihydropapaverine/(S)-tetrahydrocolumbamine	1.67 ± 0.64	**	−0.67
furcatin	1.66 ± 0.55	*	−0.40
1-[18-hydroxyoeoyl]-2-[18-hydroxy-linoleoyl]-sn-glycerol	1.66 ± 0.51	*	−9.12
1,2-dioleoylglycerol	1.66 ± 0.54	n.s.	−9.53
(R)-S-lactoylglutathione	1.65 ± 0.59	n.s.	−4.00
(3Z,6Z)-nonadienal	1.65 ± 0.64	n.s.	−2.09
Fe(II)-nicotianamine	1.64 ± 0.56	*	−0.77
(-)-4’-demethyl-deoxypodophyllotoxin/S-adenosyl-L-homocysteine	1.64 ± 0.67	**	−0.59
quercetin-3-rhamnoside-7-rhamnoside/vicenin-2/isovitexin 2″-O-β-D-glucoside/vitexin 2″-O-β-D-glucoside/rhamnosylisoorientin	1.64 ± 0.59	*	−0.48
A phenylpropanoid	1.64 ± 0.71	n.s.	11.57
A phenylpropanoid conjugate	1.63 ± 0.60	*	−0.48
L-dopachrome	1.63 ± 0.97	**	1.97
protoheme	1.63 ± 0.62	**	−1.21
pelargonidin 3-O-β-D-p-coumaroylglucoside	1.62 ± 0.62	*	−0.47
D-tryptophan	1.62 ± 0.73	n.s.	9.09
(4S)-2,3-dehydroleucopelargonidin/1,3,8-trihydroxy-2-methoxyanthraquinone/scutellarein/aureusidin	1.61 ± 0.71	*	−0.43
cyanidin 3-O-β-D-p-coumaroylglucoside/cyanidin 3-(p-coumaroyl)-glucoside/pelargonidin 3-O-β-D-caffeoylglucoside	1.60 ± 0.78	*	−1.30
N-vanillate-L-glutamate	1.60 ± 0.71	**	−1.56
(2S)-eriodictyol/2-hydroxynaringenin/1-(4-hydroxyphenyl)-3-(2,4,6-trihydroxyphenyl)propane-1,3-dione/dalbergioidin	1.60 ± 0.71	*	−0.43
7-(methylsulfanyl)heptyl-desulfoglucosinolate	1.60 ± 0.71	**	0.49
1-aminocyclopropane-1-carboxylate/azetidine-2-carboxylate	1.60 ± 0.77	n.s.	n.s.
(-)-isopiperitenone	1.60 ± 0.56	*	2.08
(-)-yatein	1.60 ± 0.73	**	0.47
1,2-di-O-sinapoyl-β-D-glucose/7-O-methylvitexin 2″-O-β-L-rhamnoside	1.60 ± 0.61	n.s.	−11.11
TRIBOA-β-D-glucoside	1.59 ± 0.85	**	−0.72
(4S)-2,3-dehydro-leucocyanidin/2-hydroxyeriodictyol/cis-12-sulfojasmonate	1.58 ± 0.71	*	−0.42
N-hydroxyhomomethionine	1.58 ± 1.19	*	−0.46
galactopinitol	1.58 ± 0.76	n.s.	−8.41
scopolamine	1.57 ± 0.49	*	9.38
cyanidin-3-O-rutinoside-5-O-β-D-glucoside	1.57 ± 1.02	*	−0.68
(22R,23R)-28-homocastasterone	1.57 ± 0.33	*	−8.88
acenaphthenequinone	1.56 ± 0.91	*	−9.78
4-coumaroylhexanoylmethane	1.56 ± 1.13	n.s.	−10.56
2-hydroxyferulate/2-hydroxycaffeate	1.56 ± 0.52	n.s.	−9.25
hypoxanthine	1.56 ± 0.35	n.s.	4.00
7,8-dihydromonapterin	1.56 ± 0.35	n.s.	13.90
(R)-prunasin	1.56 ± 0.35	n.s.	23.85
9-[6(S),9-diamino-5,6,7,8,9-pentadeoxy-β-D-ribo-nonafuranosyl]-9H-purin-6-amine	1.55 ± 0.35	*	8.78
hydroxyechinenone	1.55 ± 0.34	n.s.	21.57
luteoforol/(+)-catechin/leucopelargonidin	1.54 ± 0.90	n.s.	−9.26
(5Z,8Z,11Z,14Z,17Z)-icosapentaenoate	1.54 ± 0.47	n.s.	−7.60
(E)-2-(1H-indol-3-yl)-1-thioacetohydroximate	1.54 ± 0.72	n.s.	n.s.
solavetivone/artemisinic aldehyde/germacra-1(10),4,11(13)-trien-12-al/(4S)-4-(5,5-dimethylcyclohex-1-en-1-yl)cyclohex-1-ene-1-carbaldehyde/zerumbone	1.53 ± 0.54	n.s.	0.24
cysteine	1.53 ± 0.99	**	−1.27
olivetolate	1.53 ± 1.06	n.s.	n.s.
17-O-acetylajmaline	1.53 ± 0.43	n.s	0.24
phenylmethanesulfenate	1.53 ± 1.00	n.s	−0.24
3-carboxy-8-(methylsulfanyl)-2-oxooctanoate	1.53 ± 0.99	n.s.	−8.76
(E)-phenylacetaldehyde oxime/2-phenylacetamide/N-benzylformamide/(Z)-phenylacetaldehyde oxime	1.52 ± 0.54	n.s.	−0.35
solasodine 3-O-β-D-glucoside	1.52 ± 0.66	*	−2.57
6,8-dihydroxypurine	1.51 ± 0.53	n.s.	12.65
D,L-α-methylphosphinothricin	1.50 ± 0.49	n.s.	−4.12
5-[[4-methoxy-3-(phenylmethoxy)phenyl]methyl]-2,4-pyrimidinediamine	1.50 ± 1.04	*	0.88
cyclo-dopa 5-O-glucoside	1.50 ± 0.52	**	−9.12
(E)-8-(methylsulfanyl)octanal oxime	1.50 ± 0.52	n.s.	2.33
6-amino-2-oxohexanoate	1.50 ± 0.58	n.s.	−9.18
adenosine 3′,5′-bisphosphate/GDP group	1.50 ± 0.53	n.s.	n.s.
avenastenone	1.48 ± 0.76	n.s.	n.s.
3-chlorodiaminopimelate	1.48 ± 0.83	*	−0.36
3-hydroxylubimin	1.48 ± 0.92	n.s.	n.s.
(S)-4-hydroxymandelonitrile/(R)-4-hydroxymandelonitrile/2-formylaminobenzaldehyde/3-hydroxyindolin-2-one/5,6-dihydroxyindole/p-hydroxymandelonitrile/	1.48 ± 0.87	n.s.	−7.84
dihydroconiferyl alcohol/iridotrial	1.48 ± 0.58	*	0.75
1-18:3-2-18:3-digalactosyldiacylglycerol	1.47 ± 0.48	n.s.	−0.17
Se-methyl-Se-L-methionine	1.46 ± 0.70	n.s.	−6.46
(12Z,15Z)-9,10-dihydroxyoctadeca-12,15-dienoate/HPODE/(9Z,12Z)-15,16-dihydroxyoctadeca-9,12-dienoate/(9Z,15Z)-12,13-dihydroxyoctadeca-9,15-dienoate/9,10-12,13-diepoxyoctadecanoate	1.46 ± 0.53	n.s.	−3.01
benzyl alcohol 6-O-β-D-xylopyranosyl-β-D-glucopyranoside	1.46 ± 0.60	n.s.	−3.59
4-hydroxy-5-methyl-2-propyl-3(2H)-furanone	1.46 ± 0.63	n.s.	6.86
S-adenosyl 3-(methylsulfanyl)propylamine	1.45 ± 0.22	n.s.	−5.26
gibberellin A25	1.45 ± 1.23	n.s.	n.s.
2-hydroxysuccinamate/aspartate	1.45 ± 0.48	n.s.	n.s.
1-18:3-2-16:3-monogalactosyldiacylglycerol	1.44 ± 0.39	n.s.	−0.17
3-hydroxy-9-apo-δ-caroten-9-one	1.44 ± 0.52	*	8.23
(S)-2-amino-6-oxohexanoate	1.43 ± 0.28	n.s.	−0.10
(3S,5R,6R)-3,5-dihydroxy-6,7-didehydro-5,6-dihydro-12′-apo-β-caroten-12′-al	1.43 ± 0.65	*	0.45
tricoumaroyl spermidine	1.43 ± 0.81	*	7.62
CMP group	1.43 ± 0.63	n.s.	−6.18
2-C-methyl-D-erythritol-2,4-cyclodiphosphate	1.43 ± 0.69	n.s.	n.s.
D-galactosylononitol	1.42 ± 1.02	n.s.	−9.05
(indol-3-yl)acetyl-L-glutamine	1.42 ± 0.58	*	−2.57
2-carboxy-1,4-naphthoquinone	1.42 ± 1.06	*	−0.43
artemisinin	1.42 ± 0.56	n.s.	12.07
enol-oxaloacetate/oxaloacetate	1.42 ± 0.99	n.s.	−0.37
hemigossypol-6-methyl ether	1.42 ± 0.78	n.s.	n.s.
A terpenoid	1.41 ± 0.65	n.s.	6.49
chelirubine	1.41 ± 0.75	n.s.	−2.27
tartrate	1.41 ± 1.03	n.s.	−0.37
palmitoleate	1.41 ± 0.63	n.s.	6.00
1-heptanal	1.40 ± 0.76	n.s.	−6.31
16,17-dihydro-16α,17-dihydroxy gibberellin A4	1.40 ± 0.56	n.s.	4.88

**Table 2 plants-11-01829-t002:** Root metabolic markers, according to their VIP score ± standard deviation (STD), related to the OPLS-DA model in Figure 2c (effect of UV treatment). Compounds are provided with *p* value and their Log_2_FC calculated in respect to control samples. * *p* ≤ 0.05, ** *p* < 0.01, *** *p* < 0.001, n.s., not significant.

Compound	VIP Score ± STD	*p* Value	Log_2_ FC
vanillylamine	2.49 ± 0.60	n.s.	−0.08
10-deacetyl-2-debenzoylbaccatin III	2.41 ± 0.26	***	12.13
A pentose	2.40 ± 0.56	***	13.43
gardenin B/dalnigrein/3,7,3′,4′-tetramethylquercetin	2.39 ± 0.60	n.s.	4.00
heliocide B1	2.28 ± 0.31	***	2.46
N-methylanthranilate	2.16 ± 0.71	***	12.27
L-α-(methylenecyclopropyl)-glycine	2.10 ± 0.83	***	12.89
5,10-methylenetetrahydropteroyl mono-L-glutamate	2.10 ± 0.58	***	14.55
1,2-dehydroreticuline/(S)-corytuberine	2.06 ± 0.32	n.s.	11.07
1-hydroxycumene	2.03 ± 0.62	n.s.	0.98
cyclogutamate	2.01 ± 0.41	n.s.	14.83
all-trans-hexaprenyl diphosphate	1.96 ± 1.19	***	13.26
syn-copalyl diphosphate	1.93 ±0.56	n.s.	n.s.
salutaridine	1.90 ± 0.69	n.s.	3.66
quercetin 3-O-(4\-O-p-coumaroyl)-glucoside″	1.90 ± 0.85	n.s.	n.s.
geranate/(1R)-trans-chrysanthemate/N-hydroxy-L-valine	1.86 ± 0.35	*	0.56
(+)-carvone/(-)-isopiperitenone/(-)-carvone/4-isopropylbenzyl alcohol/(4S)-perillyl aldehyde/(+)-menthofuran	1.86 ± 0.37	*	0.56
(+)-bornane-2,5-dione	1.86 ± 0.46	n.s.	2.77
O-sinapoylglucarolactone	1.83 ± 1.15	*	0.49
butyl propanoate	1.82 ± 0.52	n.s.	−9.14
quercetin 3-O-(-O-p-coumaroyl)-glucoside	1.82 ± 0.90	n.s.	14.77
2,4-diamino-6-methyl-5,3′-(3-nitrophenoxy)prop-1′-yloxypyrimidine	1.78 ± 1.15	n.s.	n.s.
robustaquinone F	1.77 ± 1.06	**	0.76
(3-hydroxy-2-oxindol-3-yl)acetyl-L-aspartate	1.77 ± 1.07	**	0.77
cyanidin 3-O-β-D-caffeoylglucoside	1.76 ± 1.04	**	0.77
kaempferol 3-O-β-D-glucosylgalactoside/kaempferol 3-O-β-D-glucosyl-(1->2)-β-D-glucoside/	1.76 ± 1.04	**	0.77
cyanidin 3-O-β-D-p-coumaroylglucoside/cyanidin 3-(p-coumaroyl)-glucoside/pelargonidin 3-O-β-D-caffeoylglucoside	1.76 ± 1.05	**	0.79
4-(β-D-glucosyloxy)benzoate	1.76 ± 0.85	**	10.55
α-hydroxyheme	1.76 ± 0.98	n.s.	2.90
scopolamine	1.73 ± 0.64	n.s.	−2.15
5-oxooctanal	1.72 ± 0.94	*	−0.47
22-oxo-docosanoyl-CoA/3-oxobehenoyl-CoA	1.71 ± 0.67	n.s.	−9.97
L-nicotianamine	1.70 ± 1.00	***	15.39
soyasaponin I	1.68 ± 0.54	n.s.	−0.40
epoxypheophorbide a	1.68 ± 0.94	n.s.	9.89
heliannuol C	1.66 ± 1.09	n.s.	−3.30
1-α-linolenoyl-2-α-linolenoyl-phosphatidylcholine	1.66 ± 0.81	n.s.	−0.54
presqualene diphosphate	1.65 ± 1.05	n.s.	12.64
iridotrial/dihydroconiferyl alcohol	1.65 ± 0.67	n.s.	2.67
ricinine	1.65 ± 0.67	n.s.	2.67
conhydrinone/hygrine/	1.64 ± 0.58	n.s.	−1.92
5-hydroxy-γ-coniceine	1.61 ± 0.58	n.s.	−3.51
2′-hydroxypseudobaptigenin	1.61 ± 1.00	n.s.	−0.10
rhamnetin	1.60 ± 1.01	n.s.	−0.11
grasshopper ketone	1.59 ± 1.25	n.s.	−0.58
(E)-1-(L-cysteinylglycin-S-yl)-N-hydroxy-ω-(methylsulfanyl)pentan-1-imine	1.58 ± 0.51	n.s.	8.29
hydroxy-β-zeacarotene	1.58 ± 1.32	n.s.	−6.89
β-D-glucosyl crocetin	1.58 ± 0.49	n.s.	9.40
N-acetyl-α-D-galactosamine 1-phosphate/N-acetyl-α-D-glucosamine 1-phosphate	1.58 ± 0.89	n.s.	−0.08
curcumin 4′-O-β-D-gentiotrioside/curcumin 4′-O-β-D-gentiobiosyl 4″-O-β-D-glucoside	1.57 ± 1.25	n.s.	7.27
1-palmitoyl-2-vernoloyl-phosphatidylcholine	1.56 ± 0.70	n.s.	−1.99
4α-carboxy-5α-cholesta-7,24-dien-3β-ol	1.56 ± 0.74	*	−1.24
1-18:1-2-16:0-monogalactosyldiacylglycerol	1.55 ± 0.86	n.s.	−0.39
munjistin	1.55 ± 1.14	n.s.	−0.14
salicylate 2-O-β-D-glucoside	1.55 ± 0.95	*	−1.75
glutathione disulfide	1.54 ± 1.06	n.s.	1.51
1-18:0-2-18:3-phosphatidylethanolamine	1.54 ± 1.05	n.s.	−1.67
1-18:3-2-16:0-digalactosyldiacylglycerol/1-16:0-2-18:3-digalactosyldiacylglycerol	1.54 ± 1.16	n.s.	−0.90
N-(4-aminobenzoyl)-L-glutamate	1.54 ± 1.03	*	8.41
aloesone	1.54 ± 0.49	n.s.	13.60
A phenylpropanoid	1.53 ± 0.98	n.s.	−0.08
1-18:2-2-18:2-monogalactosyldiacylglycerol	1.52 ± 0.99	*	1.79
ginsenoside Rg1	1.52 ± 0.26	n.s.	−1.74
7,8-diaminopelargonate	1.52 ± 0.85	n.s.	−2.56
3β-hydroxy-β-cyclocitral/(6E)-8-hydroxygeranial/(6E)-8-oxogeraniol	1.52 ± 0.71	*	1.22
2,6-diaminopurine/heptanoate	1.51 ± 0.37	n.s.	−0.19
stevioside/rebaudioside B	1.51 ± 0.88	n.s.	−7.24
ajmaline	1.51 ± 0.94	n.s.	−8.37
pheophorbide b	1.51 ± 0.86	n.s.	−6.15
1-palmitoyl-2-linoleoyl-phosphatidylcholine	1.50 ± 0.94	n.s.	−0.31
22-hydroxydocosanoate	1.49 ± 1.10	n.s.	−0.39
(E)-1-(L-cysteinylglycin-S-yl)-N-hydroxy-ω-(methylsulfanyl)heptan-1-imine/sinapoyl-(S)-malate	1.47 ± 0.30	n.s.	−6.58
eupatolin	1.47 ± 1.01	n.s.	0.42
deoxyhumulone	1.47 ± 1.29	n.s.	−0.54
(S)-scoulerine/(S)-coreximine	1.47 ± 0.67	n.s.	0.28
phosphopantetheine	1.47 ± 0.30	n.s.	−6.63
delphinidin 3-O-rutinoside-7-O-glucoside/quercetin 3-O-rhamnosyl(1->2)glucoside-7-O-rhamnoside/quercetin 3-O-gentiobioside-7-O-rhamnoside/kaempferol 3-O-β-D-glucosyl-(1->2)-glucosyl-(1->2)-β-D-glucoside	1.47 ± 0.56	n.s.	4.44
cycloheptadienyl/sinapate	1.47 ± 0.96	**	−10.73
4α-carboxy-5α-cholesta-8,24-dien-3β-ol	1.46 ± 0.78	n.s.	−3.30
1-deoxy-2,3-hexodiulose-6-phosphate	1.46 ± 1.02	n.s.	−0.07
A saccharide	1.46 ± 0.98	n.s.	−0.07
(indol-3-yl)acetyl-L-proline	1.46 ± 0.57	n.s.	−6.48
catharanthine	1.46 ± 0.95	n.s.	−0.44
geranyl β-D-glucopyranoside	1.46 ± 0.89	*	0.63
ajmaline-x	1.46 ± 1.17	n.s.	n.s.
6-methoxypodophyllotoxin	1.45 ± 1.14	n.s.	3.41
p-coumaroylserotonin	1.45 ± 0.49	n.s.	8.25
(2E,4E,6E)-4-methylocta-2,4,6-trienedial	1.45 ± 0.39	n.s.	−0.18
(-)-medicarpin-3-O-glucoside	1.45 ± 0.26	n.s.	5.54
3-oxocerotoyl-CoA	1.45 ± 1.05	n.s.	−7.81
1,3,5-trimethoxybenzene	1.45 ± 0.39	n.s.	−0.18
1-linoleoyl-2-palmitoyl-phosphatidylglycerol	1.45 ± 0.90	*	−1.29
1-stearoyl-sn-glycerol 3-phosphate	1.44 ± 0.64	n.s.	−0.17
β-citraurin	1.44 ± 0.62	n.s.	−0.19
myristate	1.44 ± 0.95	n.s.	−0.32
phosphocholine	1.43 ± 1.16	n.s.	−5.00
tetramethylmyricetin/tetramethylquercetagetin	1.43 ± 0.31	n.s.	−5.43
norbelladine	1.43 ± 0.98	n.s.	7.39
cyanidin O-O-[6-O-(6-O-4-hydroxycinnamoyl-β-D-glucosyl)-2-O-β-D-xylosyl-β-D-galactoside]	1.43 ± 1.49	n.s.	0.48
cytochrome c	1.43 ± 0.64	n.s.	n.s.
gibberellin A51-catabolite	1.42 ± 0.52	n.s.	0.40
26-hydroxybrassinolide	1.42 ± 1.22	n.s.	−0.47
2-O-caffeoylglucarate	1.42 ± 0.61	n.s.	−7.41
allosamidin/N,N′,N″-triacetylchitotriose	1.41 ± 0.73	n.s.	−2.50
5-phospho-D-arabinonohydroxamate	1.41 ± 1.07	n.s.	−0.06
1,2-dipalmitoyl-phosphatidylglycerol-phosphate	1.41 ± 1.26	n.s.	−1.36
DIMBOA-β-D-glucoside	1.41 ± 0.36	n.s.	−0.76
coumarinate	1.41 ± 0.40	n.s.	−0.50
(R)-3-(4-hydroxyphenyl)lactate	1.40 ± 0.39	n.s.	−0.99
scopolin	1.40 ± 0.38	n.s.	−0.72
protochlorophyllide a	1.40 ± 0.84	n.s.	−1.85
(E)-1-(L-cysteinylglycin-S-yl)-N-hydroxy-2-(1H-indol-3-yl)ethan-1-imine	1.40 ± 0.39	n.s.	−0.67
chlorophyllide b	1.40 ± 1.29	n.s.	0.40
anthraniloyl-O-glucopyranose	1.40 ± 0.51	n.s.	−0.97
5-(methylsulfanyl)pentyl-desulfoglucosinolate	1.40 ± 0.31	n.s.	−1.30
gibberellin A12-aldehyde	1.40 ± 0.52	n.s.	8.42
isatin	1.39 ± 1.43	n.s.	−0.05
tetradecan-1-ol	1.39 ± 0.55	n.s.	0.00
1-18:3-2-16:4-monogalactosyldiacylglycerol	1.38 ± 0.98	n.s.	−2.10
(indol-3-yl)acetyl-myo-inositol L-arabinoside	1.38 ± 0.93	n.s.	0.36
cis-tuberonic acid	1.38 ± 1.31	n.s.	−6.50
(1,4)-β-xylobiose/7,8-dimethoxy-flavone/3-methylinosine	1.38 ± 0.92	n.s.	7.30
p-aminobenzoate-β-D-glucopyranosyl ester	1.38 ± 0.48	n.s.	−7.95
1-(3,4-dihydroxyphenyl)-5-hydroxy-3-decanone/3″-hydroxy-geranylhydroquinone	1.38 ± 1.50	n.s.	−0.74
phenylarsine oxide	1.37 ± 1.02	n.s.	−0.05
tyramine	1.37 ± 0.63	n.s.	−0.16
trans-tuberonic acid	1.37 ± 1.31	n.s.	−1.18
CDP-1-18:1(9Z)-2-18:1(9Z)-glycerol	1.36 ± 0.89	*	0.22
phlorisobutanophenone	1.36 ± 1.34	n.s.	5.51
16-epivellosimine	1.36 ± 1.23	n.s.	0.00
5-mercaptodeoxyuridine	1.36 ± 0.77	n.s.	−0.08
indole-3-carbinol	1.35 ± 1.06	n.s.	−6.80
capsanthin/antheraxanthin/4-hydroxyzeaxanthin	1.35 ± 0.85	n.s.	−0.64
3,4-dihydrocoumarin/4-coumaraldehyde/1-phenylpropane-1,2-dione	1.35 ± 0.68	n.s.	−0.15
dimethylsulfoniopropanoate-amine	1.34 ± 0.96	n.s.	−1.39
kaempferide 3-O-β-D-glucopyranosyl-(1->2)-O-α-L-rhamnoside	1.34 ± 1.49	n.s.	0.41
5-methyl-DL-tryptophan	1.34 ± 0.51	n.s.	n.s.
3,8-divinyl protochlorophyllide a	1.33 ± 1.50	n.s.	0.409
1-18:1-2-16:0-phosphatidate	1.33 ± 0.91	n.s.	−7.65
N-demethylvindolidine	1.32 ± 1.20	n.s.	−2.73
dihydrogeranylgeranyl diphosphate	1.32 ± 0.38	n.s.	n.s.
gibberellin A9	1.32 ± 0.99	n.s.	−0.29
18-hydroxyoleate/9,10-epoxystearate/ricinoleate	1.32 ± 0.73	n.s.	−0.36
(7Z,10Z,13Z,16Z,19Z)-docosa-7,10,13,16,19-pentenoate	1.32 ± 0.60	n.s.	−0.47
pheophytin b	1.32 ± 1.08	n.s.	−7.60
3-oxo-2-(cis-2′-pentenyl)-cyclopentane-1-butanoyl-CoA	1.32 ± 1.51	n.s.	0.29
kaempferol 3-O-(\-O-p-coumaroyl)-glucoside″/kaempferol-3-O-β-D-glucopyranoside-7-O-α-L-rhamnopyranoside	1.32 ± 1.41	n.s.	n.s.
linoleate	1.32 ± 0.71	n.s.	−0.36
α,ω-9Z-octadecenedioate	1.32 ± 0.68	n.s.	−0.51
prenyl diphosphate/isopentenyl diphosphate	1.31 ± 1.00	n.s.	−0.05
(+)-jasmonate	1.31 ± 0.91	n.s.	4.85
6-isobutyl-4-hydroxy-2-pyrone	1.31 ± 0.84	n.s.	−2.59
α-chaconine	1.31 ± 1.14	n.s.	−7.90
HPOTE/(9Z,11E,14Z)-(13S)-hydroperoxyoctadeca-(9,11,14)-trienoate	1.31 ± 0.71	n.s.	−6.35
palmitoleoyl-CoA	1.31 ± 1.60	n.s.	0.25
pheophytin a	1.31 ± 1.14	n.s.	−6.55
imidazole acetol-phosphate	1.31 ± 1.01	n.s.	−0.05
N-hydroxypentahomomethionine	1.30 ± 0.83	n.s.	−0.75
all-trans-nonaprenyl diphosphate	1.30 ± 0.99	n.s.	0.08
kauralexin B3	1.30 ± 0.70	n.s.	0.72

**Table 3 plants-11-01829-t003:** Discriminant phenolic compounds identified in leaves according to VIP associated with the OPLS-DA model reported in Figure 4a (effect of UV treatment). Phenolic compounds, classified according to their class and subclass, are provided with *p* value and their Log_2_FC calculated in respect to control samples. ** *p* < 0.01, *** *p* < 0.001, n.s., not significant.

Compound	Class	Subclass	VIP Score ± STD	*p* Value	Log_2_ FC
Cyanidin 3-O-(6″-p-coumaroyl-glucoside)/Petunidin 3-O-rutinoside/Pelargonidin 3-O-sophoroside/Cyanidin 3-O-rutinoside	Flavonoids	Anthocyanins	1.40 ± 0.42	n.s.	−0.26
Cyanidin	Flavonoids	Anthocyanins	1.38 ± 0.49	n.s.	−0.20
Cyanidin 3-O-glucosyl-rutinoside	Flavonoids	Anthocyanins	1.35 ± 0.64	n.s.	−0.49
Cyanidin 3-O-glucoside/Cyanidin 3-O-galactoside/Petunidin 3-O-arabinoside/Peonidin 3-O-arabinoside/p-Coumaric acid	Flavonoids	Anthocyanins	1.24 ± 0.90	n.s.	−6.47
Delphinidin 3-O-glucoside/Delphinidin 3-O-galactoside	Flavonoids	Anthocyanins	1.13 ± 0.40	n.s.	−6.09
Dihydroquercetin	Flavonoids	Dihydroflavonols	1.38 ± 0.50	n.s.	−0.20
(-)-Epicatechin 3-O-gallate/(+)-Catechin 3-O-gallate	Flavonoids	Flavanols	1.22 ± 0.97	n.s.	−6.54
Eriocitrin/Neoeriocitrin	Flavonoids	Flavanones	1.38 ± 0.48	n.s.	−0.26
Cirsimaritin	Flavonoids	Flavones	1.59 ± 0.76	***	0.72
Apigenin 7-O-(6″-malonyl-apiosyl-glucoside)	Flavonoids	Flavones	1.13 ± 0.39	n.s.	−0.54
Kaempferol 7-O-glucoside	Flavonoids	Flavonols	1.48 ± 0.72	n.s.	−2.51
Kaempferol 3-O-rutinoside/Chrysoeriol 7-O-apiosyl-glucoside/Apigenin 6.8-di-C-glucoside/Kaempferol 3-O-galactoside 7-O-rhamnoside/Luteolin 7-O-rutinoside	Flavonoids	Flavonols	1.41 ± 0.40	n.s.	−0.26
Kaempferol/Scutellarein/Luteolin	Flavonoids	Flavonols	1.38 ± 0.50	n.s.	−0.19
Kaempferol 3-O-glucosyl-rhamnosyl-galactoside/Quercetin 3-O-rhamnosyl-rhamnosyl-glucoside/Kaempferol 3-O-glucosyl-rhamnosyl-glucoside	Flavonoids	Flavonols	1.35 ± 0.64	n.s.	−0.49
3-Methoxysinensetin/Nobiletin	Flavonoids	Flavonols	1.14 ± 0.61	n.s.	−1.65
3-Methylcatechol/Guaiacol/4-Methylcatechol	Other polyphenols	Alkylphenols	1.79 ± 0.34	n.s.	4.94
5-Nonadecenylresorcinol	Other polyphenols	Alkylphenols	1.25 ± 0.75	n.s.	7.75
5-Heneicosenylresorcinol	Other polyphenols	Alkylphenols	1.10 ± 0.81	n.s.	−0.56
Umbelliferone/4-Hydroxycoumarin	Other polyphenols	Hydroxycoumarins	1.10 ± 1.00	n.s.	0.10
p-HPEA-EA/Ligstroside-aglycone	Other polyphenols	Tyrosols	1.53 ± 0.58	n.s.	−0.36
Ligstroside	Other polyphenols	Tyrosols	1.14 ± 0.52	n.s.	0.10
Galloyl glucose	Phenolic acids	Hydroxybenzoic acids	1.21 ± 1.28	n.s.	1.78
Gallic acid 4-O-glucoside	Phenolic acids	Hydroxybenzoic acids	1.18 ± 0.82	n.s.	0.08
Rosmarinic acid	Phenolic acids	Hydroxycinnamic acids	1.69 ± 0.64	***	−4.00
Caffeoyl aspartic acid	Phenolic acids	Hydroxycinnamic acids	1.52 ± 0.72	**	0.52
p-Coumaroyl tartaric acid	Phenolic acids	Hydroxycinnamic acids	1.44 ± 0.89	n.s.	1.85
4-Feruloylquinic acid/3-Feruloylquinic acid/5-Feruloylquinic acid	Phenolic acids	Hydroxycinnamic acids	1.38 ± 0.77	n.s.	−3.71
Ferulic acid 4-O-glucoside/Feruloyl glucose	Phenolic acids	Hydroxycinnamic acids	1.32 ± 0.65	n.s.	−0.44
Isoferulic acid/Ferulic acid	Phenolic acids	Hydroxycinnamic acids	1.30 ± 0.62	n.s.	−0.54
Caffeoyl tartaric acid	Phenolic acids	Hydroxycinnamic acids	1.28 ± 1.15	n.s.	−6.84
m-Coumaric acid/o-Coumaric acid	Phenolic acids	Hydroxycinnamic acids	1.24 ± 1.25	n.s.	−8.72
Avenanthramide 2c/Avenanthramide K	Phenolic acids	Hydroxycinnamic acids	1.18 ± 0.91	n.s.	0.82
Dihydrocaffeic acid/Syringaldehyde/Homovanillic acid	Phenolic acids	Hydroxyphenylpropanoic acids	1.20 ± 1.18	n.s.	−7.57

**Table 4 plants-11-01829-t004:** Discriminant phenolic compounds identified in roots according to VIP associated with the OPLS-DA model reported in Figure 4c (effect of UV treatment). Phenolic compounds, classified according to their class and subclass, are provided with *p* value and their Log_2_FC calculated in respect to control samples. * *p* ≤ 0.05, n.s., not significant.

Compound	Class	Subclass	VIP Score ± STD	*p* Value	Log_2_ FC
Delphinidin 3,5-O-diglucoside/Delphinidin 3-O-glucosyl-glucoside	Flavonoids	Anthocyanins	1.70 ± 0.59	n.s.	0.60
Delphinidin 3-O-(6″-p-coumaroyl-glucoside)	Flavonoids	Anthocyanins	1.70 ± 0.62	n.s.	0.64
Cyanidin 3-O-(6″-caffeoyl-glucoside)	Flavonoids	Anthocyanins	1.70 ± 0.62	n.s.	0.64
Pelargonidin 3,5-O-diglucoside	Flavonoids	Anthocyanins	1.68 ± 0.48	n.s.	8.60
Pigment A/Peonidin 3-O-(6″-p-coumaroyl-glucoside)	Flavonoids	Anthocyanins	1.51 ± 0.63	n.s.	−0.58
Peonidin 3-O-rutinoside	Flavonoids	Anthocyanins	1.17 ± 1.05	n.s.	−7.18
Prodelphinidin dimer B3	Flavonoids	Flavanols	1.70 ± 0.60	n.s.	0.61
Gardenin B	Flavonoids	Flavones	2.19 ± 0.86	n.s.	4.00
Nepetin/Isorhamnetin/Rhamnetin	Flavonoids	Flavones	1.41 ± 0.75	n.s.	−0.28
Apigenin 7-O-(6″-malonyl-apiosyl-glucoside)	Flavonoids	Flavones	1.31 ± 0.79	n.s.	−0.99
Quercetin 3-O-galactoside 7-O-rhamnoside/Kaempferol 3-O-sophoroside/Quercetin 3-O-rutinoside/Kaempferol 3,7-O-diglucoside/Quercetin 3-O-rhamnosyl-galactoside	Flavonoids	Flavonols	1.72 ± 0.56	n.s.	3.03
Quercetin/Morin/p-Coumaroyl malic acid/6-Hydroxyluteolin	Flavonoids	Flavonols	1.70 ± 0.62	n.s.	0.58
4-Vinylguaiacol/3-Methoxyacetophenone	Other polyphenols	Alkylmethoxyphenols	1.40 ± 0.45	n.s.	−0.36
p-Anisaldehyde	Other polyphenols	Hydroxybenzaldehydes	1.14 ± 1.06	n.s.	−0.83
Thymol/Carvacrol	Other polyphenols	Phenolic terpenes	1.67 ± 0.56	n.s.	0.39
Hydroxytyrosol 4-O-glucoside	Other polyphenols	Tyrosols	1.24 ± 0.73	n.s.	−1.00
3,4-DHPEA-AC	Other polyphenols	Tyrosols	1.10 ± 1.09	n.s.	5.34
4-Hydroxybenzoic acid 4-O-glucoside	Phenolic acids	Hydroxybenzoic acids	1.16 ± 0.53	n.s.	0.018
Sinapic acid	Phenolic acids	Hydroxycinnamic acids	1.54 ± 1.26	n.s.	8.77
p-Coumaroyl glycolic acid	Phenolic acids	Hydroxycinnamic acids	1.44 ± 0.63	*	−10.91
5-Caffeoylquinic acid/3-Caffeoylquinic acid/4-Caffeoylquinic acid	Phenolic acids	Hydroxycinnamic acids	1.25 ± 0.19	n.s.	9.47
p-Coumaric acid	Phenolic acids	Hydroxycinnamic acids	1.21 ± 0.25	n.s.	5.25
Caffeic acid	Phenolic acids	Hydroxycinnamic acids	1.11 ± 0.98	n.s.	5.34
Homoveratric acid	Phenolic acids	Hydroxyphenylacetic acids	1.10 ± 1.09	n.s.	−0.36
d-Viniferin/e-Viniferin/Pallidol	Stilbenes	Stilbenes	1.19 ± 0.28	n.s.	0.57

**Table 5 plants-11-01829-t005:** List of the phenolic compounds in UV-treated leaves, selected by combining analysis of variance and fold-change into volcano plot (Benjamini–Hochberg multiple testing correction, *p* < 0.05; fold-change cut-off = 1.2). The symbols “↑” and “↓” mean increased and decreased accumulation, respectively.

Sample	Phenolic Class	Phenolic Subclass	Marker	Log_2_ FC	Accumulation
UV-11d	Other polyphenols	Tyrosols	Oleoside 11-methylester	17.42	↑
	Phenolic acids	Hydroxycinnamic acids	Rosmarinic acid	−19.09	↓
UV-rec	Flavonoids	Flavones	Cirsimaritin	17.68	↑
		Flavonols	6,8-Dihydroxykaempferol/Myricetin	13.85	↑
		Flavonols/Flavonols	Nepetin/Isorhamnetin/Rhamnetin	16.44	↑
		Flavones	Tetramethylscutellarein	−15.90	↓
	Lignans	-	Dimethylmatairesinol	19.12	↑
		Lignans/Hydroxybenzoic acids	Protocatechuic aldehyde	−18.90	↓
		Lignans/Hydroxybenzoic acids	Sesamol/4-Hydroxybenzoic acid/2-Hydroxybenzoic acid	−18.92	↓
	Phenolic acids	Hydroxybenzoic acids	3-Hydroxybenzoic acid	−18.90	↓
			Gallic acid 4-O-glucoside	−8.90	↓
			Galloyl glucose	−8.90	↓
		Hydroxycinnamic acids	Caffeoyl aspartic acid	18.02	↑
			Ferulic acid 4-O-glucoside/Feruloyl glucose	−17.44	↓
			m-Coumaric acid/o-Coumaric acid	−15.02	↓
			p-Coumaric acid	15.95	↑
			p-Coumaroyl tartaric acid	18.59	↑
			Rosmarinic acid	−18.83	↓
		Hydroxyphenylpropanoic acids	Dihydro-p-coumaric acid/Methoxyphenylacetic acid	−12.05	↓
	Other polyphenols	Alkylmethoxyphenols/Hydroxybenzoketones	4-Vinylguaiacol/3-Methoxyacetophenone	12.67	↑
		Alkylphenols	4-Vinylphenol	20.91	↑
		Hydroxybenzaldehydes	p-Anisaldehyde	−12.43	↓
		Hydroxycoumarins	Coumarin	14.96	↑
		Tyrosol	3,4-DHPEA-EDA	−15.68	↓
		Tyrosols/Alkylphenols	Tyrosol/4-Ethylcatechol	20.92	↑

**Table 6 plants-11-01829-t006:** List of the phenolic compounds in UV-treated roots, selected by combining analysis of variance and fold-change into volcano plot (Benjamini–Hochberg multiple testing correction, *p* < 0.05; fold-change cut-off = 1.2). The symbols “↑” and “↓” mean increased and decreased accumulation, respectively.

Sample	Phenolic Class	Phenolic Subclass	Marker	Log_2_ FC	Accumulation
UV-11d	Flavonoids	Flavones	Gardenin B	20.62	↑
UV-rec	Flavonoids	Flavones	Gardenin B	21.69	↑
			Jaceidin 4′-O-glucuronide	−15.19	↓
	Phenolic acids	Hydroxycinnamic acids	5-Caffeoylquinic acid/3-Caffeoylquinic acid/4-Caffeoylquinic acid	−21.25	↓
			p-Coumaroyl glycolic acid	−13.02	↓
			p-Coumaric acid	−19.58	↓
			Sinapic acid	4.45	↑
	Lignans	-	Pinoresinol/Matairesinol	−8.43	↓
	Phenolic acids/Other polyphenols/Phenolic acids	Hydroxyphenylpropanoic acids/Hydroxybenzaldehydes/Hydroxyphenylacetic acids	Dihydrocaffeic acid/Syringaldehyde/Homovanillic acid	16.32	↑
	Other polyphenols	Alkylmethoxyphenols/Hydroxybenzoketones	4-Vinylguaiacol/3-Methoxyacetophenone	−19.39	↓
		Alkylphenols	5-Pentadecylresorcinol	−18.23	↓
			4-Vinylphenol	−21.38	↓
		Hydroxyphenylpropenes	Acetyl eugenol	−19.07	↓
		Phenolic terpenes	Carnosic acid	−17.60	↓
		Stilbene	Resveratrol	−18.66	↓
		Tyrosols	Hydroxytyrosol 4-O-glucoside	19.83	↑
		Tyrosols/Alkylphenols	Tyrosol/4-Ethylcatechol	−21.37	↓

## Data Availability

The datasets presented in this study are available in Appendix A.

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
