# Peer review of "Foliar and Root Comparative Metabolomics and Phenolic Profiling of Micro-Tom Tomato (Solanum lycopersicum L.) Plants Associated with a Gene Expression Analysis in Response to Short Daily UV Treatments"

_plants, 2022, doi:10.3390/plants11141829_

Round 1

Reviewer 1 Report

Manuscript “Foliar and root ……… UV treatments” deals with the metabolite profiling of tomato under UV stress treatment and associated responsive gens.  

I have few concerns about the study:—

1.       UHPLC-ESI-QTOF-MS analysis is quantitative or qualitative?

2.       If quantitative, and how you have calculated concentration?

3.       If qualitative, how you have calculated fold change?

4.       The whole study is untargeted metabolomics, authors are advice to estimate quantities of the at least marker metabolites under normal and stressed conditions as per table 5?

5.       Presence of some metabolites are very surprisingly, for example Ricinine (table 2) which is a toxic alkaloid specific to castor plant; Eupatolin which is a chemical compound; Isatin an organic compound; so on

6.       Authors, please recheck the all metabolites, removed which are synthetic or irrelevant?

7.       No information that how they calculated relative fold expression in real time PCR?

8.       Provide, if possible, melt curve analysis to conform primer specificity.

Author Response

Dear Reviewer, please find attached the file with our point-by-point responses to your comments.

Reviewer 2 Report

The identification of metabolites in this paper is questionable. I immediately noticed a very large number of metabolites, 2800 and 3000, so I had to check the identification data contained in the supplementary data. Namely, a closer look at the document "table S2_dataset", I found a significant number of errors. For example, leaf sheets in the rows 2120, 2132, 2149, 2250, 2270, and 2293 contain compounds that are structural isomers but have identical experimental data. In attached document you will find a table with all this data: same abundance, same annotation and Composite Spectrum. It seems that there is only one compound, not 6 as you stated, because the data for all 6 are the same. The same abundances were used as variables, so there are no 6 variables, but one in this case. The actual number of metabolites appears to be significantly lower.

The biggest problem with identification is the Composite Spectrum, which should contain the exact masses of metabolites. What do these numbers represent: (613.17346, 6758.3853) (614.1715, 1498.9529)? In the attach you will find the structural formula of the compound in row 2293, the exact mass of this compound in the positive mode is around 627.16 m/z, and in the negative around 625.14 m/z (not 613 or 614 m/z).

This paper is based on statistical analysis of data  obtained from UHPLC ESI TOF MS analysis, which may not be accurate. I suggest that the authors check the table with metabolites, insert MS2 data, because that will help a lot with identification. Also, I suggest that it is better to reduce the number of metabolites, but to identify them more precisely, without repeating the same compounds, and with more data (retention times, MS2, relative abundance of MS2 fragments).

At the moment, before the further and more detailed consideration of the manuscript, I think that it is necessary to present the suggestions to the authors. I think the topic of the research itself is interesting and can attract readers, but the manuscript requires a lot of refinement, both technical and substantive nature.

Author Response

(The authors gave the same response as above.)

Reviewer 3 Report

It is certainly impossible not to note the enormous amount of experimental data obtained in the course of the work. It is also worth mentioning the very good statistical presentation of the results. Such presentation of the data facilitates their perception. However, the work has some shortcomings, the elimination of which would significantly increase the value of the work. 

 These are not deficiencies in the design and execution of experiments, but rather in their development and discussion. First, I do not refer to previous similar results obtained with fruit. A comparison of such data would give a complete picture of the plant under stress and provide a holistic view of the changes in the plant. And such a view, in turn, will allow for bolder research hypotheses.

 The results obtained are not very informative, to say the least - changes in phenolic compounds are what one should expect - and only a proper interpretation of the data would add value to the novelty. In a word, the discussion of the results should be expanded.

 The authors note changes in the content of fatty acids (PUFAs?) and discuss them only in passing (the structure of the cell wall), without mentioning, for example, the possibility of the formation of free fatty acids. They do not mention the possible role of phenolic compounds in interrupting the oxidative process (which groups of compounds might be involved).

 One should consider the mechanism of changes in the root and stem (p. 19, lines 284-285). The explanation of "indirectly" is not sufficient. Whether it is the effect of equilibrium synthesis/decay or detour of substrates, or perhaps the uptake of products and transport to other organs of the plant - this would require a look at the plant as a whole.

 Without such a view, the work is not worth much from a scientific point of view. It is just a corner of artisanal, undoubtedly hard work.

Author Response

(The authors gave the same response as above.)

Round 2

Reviewer 1 Report

Revision is satisfactory

Author Response

We thank the reviewer.

Author Response

Dear reviewer, 

please find attached our reply to your observation.

Reviewer 3 Report

The manuscript in its present form may be accepted.

Author Response

We thank the reviewer.